# AttrSeg: Open-Vocabulary Semantic Segmentation via Attribute Decomposition-Aggregation

**Chaofan Ma**[1], **Yuhuan Yang**[1], **Chen Ju**[1], **Fei Zhang**[1], **Ya Zhang**[1,2], **Yanfeng Wang**[1,2✉]

[1] Coop. Medianet Innovation Center, Shanghai Jiao Tong University
[2] Shanghai AI Laboratory

{chaofanma, yangyuhuan, ju_chen, ferenas, ya_zhang, wangyanfeng622}@sjtu.edu.cn

## Abstract

Open-vocabulary semantic segmentation is a challenging task that requires segmenting novel object categories at inference time. Recent works explore vision-language pre-training to handle this task, but suffer from unrealistic assumptions in practical scenarios, *i.e.*, low-quality textual category names. For example, this paradigm assumes that new textual categories will be accurately and completely provided, and exist in lexicons during pre-training. However, exceptions often happen when meet with ambiguity for brief or incomplete names, new words that are not present in the pre-trained lexicons, and difficult-to-describe categories for users. To address these issues, this work proposes a novel *attribute decomposition-aggregation* framework, **AttrSeg**, inspired by human cognition in understanding new concepts. Specifically, in the *decomposition* stage, we decouple class names into diverse attribute descriptions to complement semantic contexts from multiple perspectives. Two attribute construction strategies are designed: using large language models for common categories, and involving manually labelling for human-invented categories. In the *aggregation* stage, we group diverse attributes into an integrated global description, to form a discriminative classifier that distinguishes the target object from others. One hierarchical aggregation architecture is further proposed to achieve multi-level aggregations, leveraging the meticulously designed clustering module. The final results are obtained by computing the similarity between aggregated attributes and images embeddings. To evaluate the effectiveness, we annotate three types of datasets with attribute descriptions, and conduct extensive experiments and ablation studies. The results show the superior performance of attribute decomposition-aggregation. We refer readers to the latest arXiv version at https://arxiv.org/abs/2309.00096.

## 1 Introduction

Semantic segmentation is one of the fundamental tasks in computer vision that involves partitioning an image into some semantically meaningful regions. Despite great progress has been made, existing research has mainly focuses on closed-set scenarios, where object categories remain constant during training and inference stages [50, 24]. This assumption is an oversimplification of real-life and limits its practical application. Another line of research considers a more challenging problem, that requires the vision system to handle a broader range of categories, including novel (unseen) categories during inference. This problem is referred as open-vocabulary semantic segmentation (OVSS).

To handle OVSS, vision-language pre-training (VLP) paradigm [42, 17, 49] provides a preliminary but popular idea. By leveraging language as an internal representation for visual recognition, segmentation is formulated as a similarity between a category's textual representation and pixel-level visual representation. Following on this paradigm, recent works focus on minor improvements, *e.g.*,

37th Conference on Neural Information Processing Systems (NeurIPS 2023).

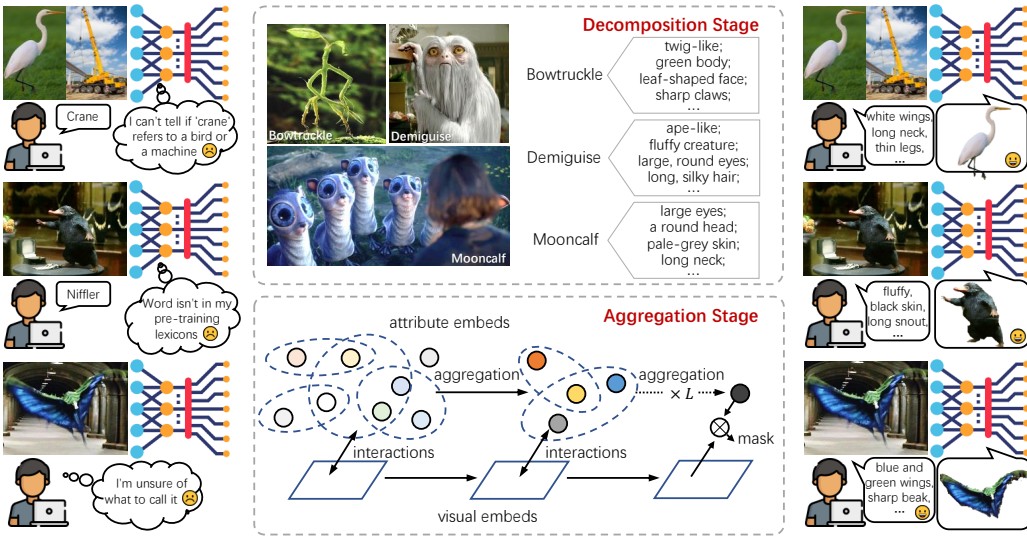

Figure 1: **Left**: Open-vocabulary semantic segmentation (OVSS) assumes the given new textual categories are accurate, complete, and exist in pre-trained lexicons. However, in real-life situations, practical uses are limited due to textual ambiguity, neologisms, and unnameability. **Middle**: We propose a novel *attribute decomposition-aggregation* framework where vanilla class names are first *decomposed* into various attribute descriptions (*decomposition stage*), and then, different attribute representations are *aggregated* hierarchically into a final class representation for further segmentations (*aggregation stage*). **Right**: Our framework successfully addresses the aforementioned issues and facilitates more practical applications of OVSS in real-world scenarios.

explore to better align vision-language modalities [52, 25, 32]. Although promising, these researches all maintain one unrealistic assumption in real-world scenarios, *i.e.*, the given new textual categories are accurate and complete, and exist in the pre-trained lexicons. As a result, three main issues persist. (1) **Ambiguity**: brief or incomplete names bring lexical ambiguity, posing a great challenge for semantic discriminability. (2) **Neologisms**: new words that frequently emerge may not be present in the lexicons during vision-language pre-training, preventing pre-trained language models from interpreting their semantics, let alone aligning them with images. (3) **Unnameability**: unnamed or difficult-to-describe categories, such as specialized terms, rare animal names, or specific objects, can create a labeling problem for users, adding complexity during application. The above three issues result in low-quality category comprehension, limiting the empirical segmentation performance.

To address these issues, we turn our attention to cognitive psychology when human understanding new concepts [36, 45]. For example, if a child asks how to find a flamingo in the zoo, one can explain the process by looking for its pink feathers, long neck, and more. Then by combining these answers, a child can easily recognize a flamingo. Such answers provide detailed descriptions from multiple distinct or complementary perspectives, which we refer to as diverse "*attributes*". Compared to vanilla categories, attributes have three advantages. (1) For ambiguous categories, attributes can make up for missing context information to achieve completeness. (2) For unseen categories, they can be transformed into known attributes, easily interpreted by pre-trained language models. (3) For unnamed or indescribable categories, attributes can be used to replace in a more detailed manner. These attribute descriptions, aggregated at scale, provide a strong basis for visual recognition.

Inspired by this, we propose a novel **decomposition-aggregation framework** where vanilla class names are first *decomposed* into various attribute descriptions, and then different attribute representations are *aggregated* into a final class representation for further segmentations. Specifically, for the **decomposition stage**, our goal is to generate various attribute descriptions from coarse category names and build attributes for datasets. We propose two construction strategies: one is to generate using language models, and the other involves manually labelling. The first strategy corresponds to situations where common category names are sometimes brief or incomplete, with semantic ambiguity or insufficient discriminability. In this case, we can simply annotate attributes upon existing datasets, such as PASCAL [13, 16] and COCO [28]. The second strategy involves

a newly collected dataset called "Fantastic Beasts", which contains imaginal creatures and their invented names by humans. This dataset is used to simulate situations for new words for pre-trained vision-language models, and difficult-to-describe categories for users. In the **aggregation stage**, our aim is to combine the separate attribute pieces into an integrated global description, which then serves as a classifier to differentiate the target object from others. This stage can also be viewed as the process of combining regions reflected by different attributes into a specific one, which yields the segmentation result. Since attributes describing objects may potentially contain hierarchy, we propose a hierarchical aggregation architecture to leverage this potential. A clustering module is carefully designed to aggregate attributes from different levels, and the final mask is obtained by computing similarities between the grouped attribute embeddings and the image features.

To evaluate the significance of attribute understanding for OVSS, we annotate attribute descriptions on three types of datasets, namely, PASCAL series [13, 16, 35], COCO series [28, 8], and Fantastic Beasts. Extensive experiments demonstrate the superior performance of our attribute decomposition-aggregation framework over multiple baselines and competitors. Furthermore, we performed thorough ablation studies to dissect each stage and component, both quantitatively and qualitatively.

To sum up, our contributions lie in three folds:

- We pioneer the early exploration in leveraging only the attribute descriptions for open-vocabulary segmentation, and to achieve this end, we construct detailed attributes descriptions for two types of existing datasets and one newly collected dataset Fantastic Beasts;
- We design a novel decomposition-aggregation framework that decompose class names into attribute descriptions, and then aggregate them into a final class representation;
- We conduct thorough experiments and ablations to reveal the significance of attribute decomposition-aggregation, and our model's superior performance on all proposed datasets.

## 2   Related Work

**Vision-Language Pre-training** (VLP) aims to jointly optimize image-text embeddings with large-scale web data. Recently, some studies have further scaled up the training to form "the foundation models", *e.g.*, CLIP [42], ALIGN [17], Florence [49], and FILIP [48]. These foundation models usually contain one visual encoder and one textual encoder, which are trained using simple noise contrastive learning for powerful cross-modal alignment. They have shown promising potential in many tasks: grounding [18, 20], detection [21, 19], and segmentation [47, 31, 51, 29]. This paper uses CLIP for OVSS, but the same technique should be applicable to other foundation models as well.

**Open-Vocabulary Semantic Segmentation (OVSS)** aims to understand images in terms of categories described by textual descriptions. Pioneering works [7, 26, 15] use generative models to synthesize visual features from word embeddings of novel categories. SPNet [46] and JoEm [4] employ a mapping process that assigns each pixel and semantic word to a joint embedding space. Recently, researchers have proposed to leverage pre-trained vision-language models (VLMs) for OVSS. OpenSeg [14] aligns region-level visual features with text embedding via region-text grounding. LSeg [25] aligns pixel-level visual embeddings with the category text embedding of CLIP. Subsequent methods like Fusioner [32], Zegformer [11], OVseg [27] and CAT-Seg [10] have thoroughly investigated the open-vocabulary capability of CLIP. However, these methods heavily rely on category names, ignoring text ambiguity, neologisms, and unnameable are common in real-world scenarios. This paper designs novel framework of attribute decomposition-aggregation to tackle these issues.

**Attribute Understanding.** Visual attributes are first studied in the traditional zero-shot learning [23, 43, 22]. With the emergence of vision-language models, the attribute understanding has developed towards a more scalable, open, and practical direction. One line of research is focused on detecting and recognizing an open set of objects, along with an open set of attributes for each object [39, 40, 5, 9]. Another line of research focus on object classification by incorporating attributes as part of text prompts, which aim to evaluate the discriminative ability of VLMs with enriched text [41, 37], or to enhance interpretability and explainability of model reasoning [34, 33]. Different from above, our work investigates attribute understanding from the perspective of open-vocabulary semantic segmentation, using a decomposition-aggregation strategy.

# 3 Method

This paper considers open-vocabulary semantic segmentation (OVSS). We start by giving the preliminary in Sec. 3.1; then we introduce our attribute decomposition-aggregation framework in Sec. 3.2; decomposition stage and aggregation stage will be detailed in Sec. 3.3 and Sec. 3.4, respectively.

## 3.1 Problem Formulation & Preliminary

**Problem.** Given an image $\mathcal{I} \in \mathbb{R}^{H \times W \times 3}$, OVSS aims to train one model $\Phi(\Theta)$ that can segment the target object according to its text description $\mathcal{T}$, that is, outputting one pixel-level mask $\mathcal{M}$:

$$\mathcal{M} = \Phi_{\text{seg}}(\mathcal{I}, \mathcal{T}; \Theta) \in \{0, 1\}^{H \times W \times 1}. \tag{1}$$

Under open-vocabulary settings, training classes $\mathcal{C}_{\text{base}}$ and testing class $\mathcal{C}_{\text{novel}}$ are disjoint, *i.e.*, $\mathcal{C}_{\text{base}} \cap \mathcal{C}_{\text{novel}} = \varnothing$. During training, image-mask pairs from the base class are provided, *i.e.*, $\{(\mathcal{I}, \mathcal{M}) \sim \mathcal{C}_{\text{base}}\}$; while during testing, the model is evaluated on the disjoint novel classes, *i.e.*, $\{\mathcal{I} \sim \mathcal{C}_{\text{novel}}\}$.

**Vision-Language Paradigm.** To enable open-vocabulary capability, recent OVSS studies [52, 25, 32] embrace vision-language pre-trainings (VLPs), for their notable ability in cross-modal alignment. Specifically, regarding vanilla class names as textual descriptions, open-vocabulary segmentation can be achieved by measuring the similarity between class-level textual and pixel-level visual embeddings:

$$\mathcal{M} = \mathcal{F}_v * \mathcal{F}_t, \quad \mathcal{F}_v = \Phi_{\text{vis}}(\mathcal{I}) \in \mathbb{R}^{H \times W \times D}, \quad \mathcal{F}_t = \Phi_{\text{txt}}(\mathcal{T}) \in \mathbb{R}^{1 \times D}, \tag{2}$$

where $\Phi_{\text{vis}}$ and $\Phi_{\text{txt}}$ refer to the visual and textual encoders in VLPs. This paradigm has a fancy dream, but meets poor reality. In practice, the textual names of novel classes may potentially suffer low-quality comprehension in three aspects: (1) *Ambiguity*. Certain names exhibit lexical ambiguity, while others sometimes may incomplete due to excessive simplification. These result in a deficiency of semantic discriminability. (2) *Neologisms*. The pre-training text corpus is inevitably limited in its coverage of vocabulary, and thus may not include certain terms that have emerged as neologisms in the real world. (3) *Unnameability*. Certain categories of entities may lack a known or easily describable name for users, particularly in cases involving specialized terminology, rare or obscure animal names, etc. These issues greatly limit the use and development of open-vocabulary segmentation.

## 3.2 Attribute Decomposition-Aggregation Framework

To solve the above issues, we introduce one novel *attribute decomposition-aggregation* framework.

**Motivation.** Such textual semantic issues are caused by the low-quality category comprehension. We consider to *decompose* class name from multiple perspectives, such as color, shape, parts and material, etc. This forms informative *attribute* sets for the text stream. Treated as partial representations describing categories, (1) attributes can supplement missing information for incompleteness and ambiguity; (2) for new words, attributes can be transformed into known words, which can be easily interpreted by the pre-trained language model; (3) for unnamed or not easily describable categories, attributes can be used to describe them in a more detailed and accurate way. These attributes, when *aggregated* to a global description, can provide a strong basis for visual recognition.

**Framework Overview.** As illustrated in Fig. 2, given an image and a set of attributes descriptions (Sec. 3.3), we first obtain its visual and attribute embeddings (Sec. 3.4.1). Considering the potential hierarchy inside attributes, we suggest a hierarchical pipeline to progressively aggregate all given attributes embeddings into one specific token (Sec. 3.4.2). As the final grouped token represents all attributes' information, segmentations can be acquired by computing the similarity between this token and the visual embeddings (Sec. 3.4.3). Formally,

$$\mathcal{M} = \mathcal{F}_v * \mathcal{F}_t, \quad \mathcal{F}_v = \Phi_{\text{vis}}(\mathcal{I}) \in \mathbb{R}^{H \times W \times D}, \quad \mathcal{F}_t = \Phi_{\text{aggr}} \circ \Phi_{\text{txt}} \circ \Phi_{\text{decp}}(\mathcal{T}) \in \mathbb{R}^{1 \times D}, \tag{3}$$

where $\Phi_{\text{decp}}(\cdot)$ denotes the decomposition module that returns the set of $n$ attributes' textual descriptions of the target category. Note that, category names (and synonyms) are strictly *not* contained in this set. $\Phi_{\text{aggr}}(\cdot)$ refers to the aggregation module that groups attributes into one specific embedding.

## 3.3 Decompose: Detailed Attribute Descriptions for Class Names

In real-world scenarios, vanilla class names $\mathcal{T}$ may be coarse-grained, ambiguous, or neologisms. Unfortunately, there is a lack of existing datasets that provide detailed attribute descriptions to offer

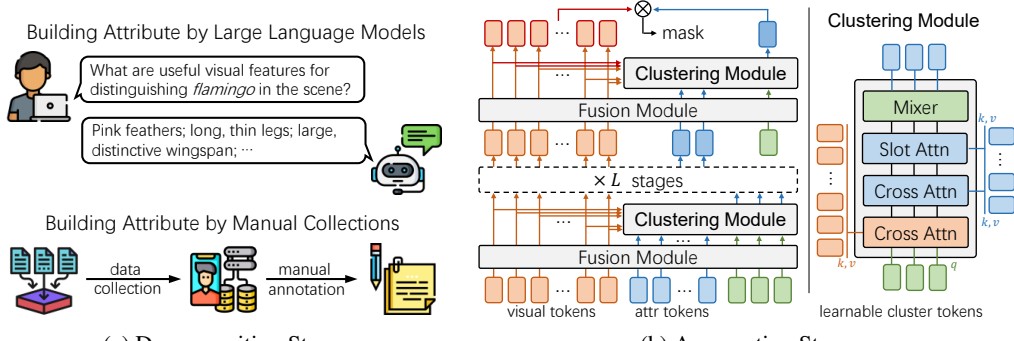

<table>
<tr><td>(a) Decomposition Stage</td><td>(b) Aggregation Stage</td></tr>
</table>

Figure 2: **Overview of Attribute Decomposition-Aggregation Framework**. **(a)** Decomposition stage aims to decouple vanilla class names into various attribute descriptions. We design two strategies to build attributes, *i.e.*, using LLMs and manual collections. **(b)** Aggregation stage aims to merge separated attribute representations into an integrated global description. We propose to hierarchically aggregate attribute tokens to one specific token in $L$ stages. Each stage alternates a fusion module and a clustering module. Masks are generated by calculating the similarity.

informative contexts. So, we here propose two strategies to construct diverse attributes $\mathcal{A}$. As shown in Fig. 2a, one involves utilizing large language models, while the other relies on manual collection.

$$\mathcal{A} = \Phi_{\text{decp}}(\mathcal{T}) = \{\text{attr}_1, \text{attr}_2, \dots, \text{attr}_n\}. \tag{4}$$

### 3.3.1 Attribute Descriptions by Large Language Models

For the cases where vanilla category names are semantically coarse or ambiguous, one promising solution is to describe attributes or contexts for better discriminability. To generate such attributes, manual writing can be time-consuming and inefficient, particularly for a large number of classes. Hence, for cost-effectiveness, we turn to large language models (LLMs) [6, 38], pre-trained on large corpora of data, showing remarkable performance of semantic understanding and text generation.

More specifically, to automatically adapt LLMs to enrich class contexts, we carefully designed a set of question templates for attribute descriptions mining from various perspectives. Taking the category "*flamingo*" as an example, we first pose multiple questions, such as "*List all attributes for distinguishing a {flamingo} in a photo*" or "*What are visual features of a {flamingo} in the image*". Then we prompt ChatGPT [38] to obtain answers of attribute descriptions, such as "*pink feathers; long neck; thin legs; large wingspan; ...*". Finally, we filter and combine answers to form an attribute set for each category. Please refer to the supplementary materials for further details.

### 3.3.2 Attribute Descriptions by Manual Collection

In addition to the above cases, there are two common cases that require attribute descriptions. One is vanilla category names are neologisms that are unseen by LLMs and VLPs; the other is when users are not familiar with an object, so they may have difficulty naming it, especially when it comes to a rare or obscure category. Given that, existing datasets typically do not include rare or obscure vocabulary, so we manually collect a dataset of human-made objects and rare categories for simulation.

The dataset is called "**Fantastic Beasts**", which consists of 20 categories of magical creatures from the film series of Fantastic Beasts [1, 2, 3]. We first scrap and filter the dataset images from the web, then organize fan volunteers of the film series to carefully annotate the paired masks and the category attributes. Since all these creatures and names are human inventions, they are unlikely to been learned by existing LLMs and VLPs. Some images, alone with their corresponding class names and attributes, are shown in Fig. 1. Please also refer to the supplementary materials for dataset information.

### 3.3.3 Discussion

(1) **Datasets**: developing high-quality attribute datasets is a crucial contribution towards advancing the practicality of OVSS. As there are currently no existing benchmarks and evaluations, an essential first step has been made that involves annotating the attributes on top of the existing datasets such as

PASCAL and COCO, as well as manually collecting a dataset, Fantastic Beasts. (2) **Why LLMs**: thanks to the capabilities of LLMs, category attributes for existing datasets can be obtained in a scalable manner. Despite not receiving any visual input during training, LLMs can successfully imitate visual attributes since they are trained on a large corpus containing descriptions with visual knowledge. (3) **Significances**: we believe that the attribute decomposition strategy, and the produced datasets, will have a great impact on the community to further promote practical uses of OVSS.

## 3.4 Aggregate: Hierarchical Fusion for Vision-Attribute Alignments

Given image $\mathcal{I}$ and attribute descriptions $\mathcal{A}$, we aim to aggregate these separate pieces of attribute embeddings *hierarchically* into one integrated global representation $\mathcal{G}$, with the help of visual information. Then, it can serve as a discriminative classifier to distinguish the target object.

$$\mathcal{G} = \Phi_{\text{aggr}}(\Phi_{\text{txt}}(\mathcal{A}); \ \Phi_{\text{vis}}(\mathcal{I})) \in \mathbb{R}^{1 \times D}. \tag{5}$$

### 3.4.1 Vision Embeddings and Attribute Embeddings

We here adopt vision-language pre-trainings [42] as encoders and mostly consider ViT-based architectures [12], due to their good performance, and flexibility for encoding different modalities.

Given an image $\mathcal{I} \in \mathbb{R}^{H \times W \times 3}$, the visual embeddings $\mathcal{V} \in \mathbb{R}^{N^v \times d}$ are extracted from the visual encoder, where $N^v$ is the number of image tokens; and $d$ is the channel dimension. The attribute embeddings $\mathcal{A} \in \mathbb{R}^{N^a \times d}$ are obtained by first feeding each attribute into the text encoder separately, then concatenating all of them, where $N^a$ is the number of attributes describing one target object.

### 3.4.2 Hierarchical Aggregation Architecture

Attributes descriptions may potentially contain hierarchy. We propose to *hierarchically* aggregate these attributes in $L$ stages, hoping to explicitly leveraging this potential, as shown in Fig. 2b.

**Overview.** Each stage alternates a *fusion module* and a *clustering module*. Specifically, the *fusion module* facilitates interaction between different modalities. Given these enriched representations, the following *clustering module* groups attribute tokens to fewer tokens. This procedure utilizes *learnable cluster tokens* as centers for clustering, and considers both visual and attribute information. Based on the similarity, these clustering centers can gather and merge all attributes tokens into specific groups.

Formally, for the $l$-th stage, we denote $N^v$ visual tokens as $\mathcal{V}_l \in \mathbb{R}^{N^v \times d}$; $N^a_l$ attribute tokens describing one target object as $\mathcal{A}_l \in \mathbb{R}^{N^a_l \times d}$; and $N^g_l$ learnable cluster tokens for aggregation as $\mathcal{G}_l \in \mathbb{R}^{N^g_l \times d}$. The fusion module fuses and enriches the information globally between $\mathcal{V}_l$, $\mathcal{A}_l$, and $\mathcal{G}_l$:

$$\mathcal{V}_l, \ \mathcal{A}_l, \ \mathcal{G}_l = \Psi^l_{\text{fuse}}(\mathcal{V}_l, \mathcal{A}_l, \mathcal{G}_l). \tag{6}$$

To avoid notation abuse, we still use the same notation for the output. After fusion, the $N^a_l$ attribute tokens $\mathcal{A}_l$ are merged and grouped to fewer $N^a_{l+1}$ ($N^a_{l+1} < N^a_l$) tokens $\mathcal{A}_{l+1}$ through clustering:

$$\mathcal{A}_{l+1} = \Psi^l_{\text{cluster}}(\mathcal{A}_l; \mathcal{V}_l, \mathcal{G}_l) \in \mathbb{R}^{N^a_{l+1} \times d}. \tag{7}$$

Note that, the attribute tokens are grouped not only based on itself, but also depending on the visual embeddings. And the number of grouped attributes $\mathcal{A}_{l+1}$ (output for this stage) is equal to the number of input learnable cluster tokens $\mathcal{G}_l$ for this stage, *i.e.*, $N^a_{l+1} = N^g_l$.

**Fusion Module.** $\Psi^l_{\text{fuse}}$ is flexible and adaptable to multiple network architectures. Here we use multiple transformer encoder layers as representatives, in order to effectively capture and propagate the long-range information of different modalities, by iteratively attending to each other.

**Clustering Module.** Learnable cluster tokens are used here to represent the clustering center for each grouping stage. It's unreasonable to aggregate solely based on attribute embeddings, as the visual information also plays a part in the segmentation. To better incorporate both modalities, the learnable clustering center first obtain the contextual information through vision and attribute cross attentions:

$$\mathcal{G}_l = \phi_{\text{cross-attn}}(q = \mathcal{G}_l, k = \mathcal{V}_l, v = \mathcal{V}_l), \quad \tilde{\mathcal{G}}_l = \phi_{\text{cross-attn}}(q = \mathcal{G}_l, k = \mathcal{A}_l, v = \mathcal{A}_l), \tag{8}$$

where $\tilde{\mathcal{G}}_l \in \mathbb{R}^{N^g_l \times d}$ is the contextual centers; $\mathcal{V}_l$ and $\mathcal{A}_l$ are visual and attribute embeddings. By exchanging information for the visual tokens and attribute tokens respectively, $\tilde{\mathcal{G}}_l$ has the knowledge of both modalities, providing a good prior for the subsequent processing.

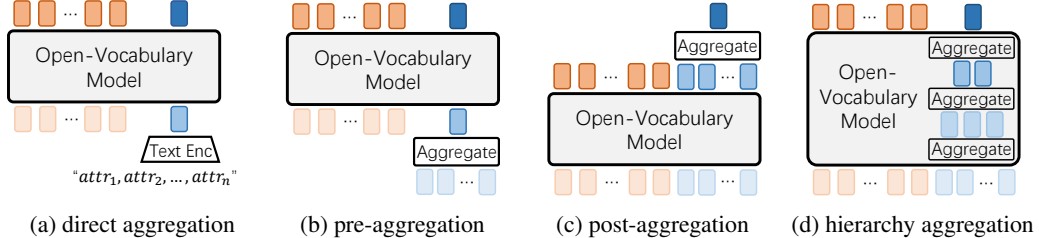

| (a) direct aggregation | (b) pre-aggregation | (c) post-aggregation | (d) hierarchy aggregation |

Figure 3: **Comparison between Various Aggregation Strategies**. The orange / blue colors represent visual / attribute tokens, respectively. Detailed discussions can be found in Sec. 3.4.4.

Next, we assign each attribute token to one of the contextual centers, and calculate the representation of newly grouped attributes $\tilde{\mathcal{A}}_l$ as output. We use slot attention [30], by seeing each cluster as a slot:

$$\tilde{\mathcal{A}}_l = \phi_{\text{slot-attn}}(q = \tilde{\mathcal{G}}_l, k = \mathcal{A}_l, v = \mathcal{A}_l), \quad \mathcal{A}_{l+1} = \phi_{\text{mixer}}(\tilde{\mathcal{A}}_l + \tilde{\mathcal{G}}_l). \tag{9}$$

The final grouped attributes $\mathcal{A}_{l+1}$ for this stage can be obtained after adding the residual to $\tilde{\mathcal{G}}_l$, and then updating and propagating the information between tokens through $\phi_{\text{mixer}}$. Here we use MLPMixer [44] with two consecutive group-wise and channel-wise MLPs.

### 3.4.3 Mask Calculation

The total $L$ stages aggregation gives the result of one specific token $\mathcal{A}_{L+1} \in \mathbb{R}^{1 \times d}$, which can be thought as condensing the collective knowledge of the provided attributes' information. Logits $\mathcal{Y}$ can be generated by computing the similarity between the final stage visual embedding $\mathcal{V}_{L+1}$ and $\mathcal{A}_{L+1}$:

$$\mathcal{Y} = \phi_{\text{sim}}(\mathcal{V}_{L+1}, \mathcal{A}_{L+1}). \tag{10}$$

The final predictions can be obtained by simply reshaping to spatial size and upsampling, then applying sigmoid with a temperature $\tau$ and thresholding.

### 3.4.4 Discussion

The fundamental concept of aggregation strategies involves identifying one single embedding that represents all attributes. As shown in Fig. 3, we present four optional designs: "**Direct**", "**Pre-**", "**Post-**", and "**Hierarchy**", which are able to be applied to all existing open-vocabulary methods. (1) "**Direct**" involves listing all attributes in one sentence, which is then directly sent to a text encoder. The aggregated embedding is obtained from the output [CLS] token, which is subsequently fed into the open-vocabulary model for further processing. (2) In contrast to "direct", the other three strategies feed attributes into the text encoder *separately*. "**Pre-**" first aggregates all attribute tokens into one token, which is then fed into the model similar to "direct". (3) "**Post-**" inputs all attribute tokens together into the open-vocabulary model. The output attribute tokens of this model are further aggregated to one token. (4) "**Hierarchy**" progressively aggregates multiple tokens into one in multiple stages, as detailed in this section. In Sec. 4.1 and Sec. 4.2, we conduct comprehensive comparisons and explore the adaptability of these strategies to existing open-vocabulary models. This analysis demonstrates the universality and effectiveness of our decomposition-aggregation motivation.

### 3.5 Training and Inference

During *training*, the visual and textual encoders are kept frozen, and we uniformly sample $N$ attributes describing the target category with replacement from the attribute set. The predicted mask is supervised by the ground truth using standard cross-entropy loss. During *inference*, the user has the flexibility to provide test images along with any number of attributes describing the object of interest. The model can then generate the corresponding segmentation mask based on this input.

## 4 Experiments

**Datasets.** We evaluate on PASCAL-$5^i$ [13, 16] COCO-$20^i$ [28] following [25, 32], and evaluate on Pascal VOC [13] and Pascal Context [35] following [11, 27, 10]. PASCAL-$5^i$ contains 20 categories

that are divided into 4 folds of 5 classes each, *i.e.*, $\{5^i\}_{i=0}^3$. COCO-20$^i$ is more challenging with 80 categories that are also divided into 4 folds, *i.e.*, $\{20^i\}_{i=0}^3$, with each fold having 20 categories. Of the four folds in the two datasets, one is used for evaluation, while the other three are used for training. PASCAL VOC is a classical dataset. We evaluate on the 1.5k validation images with 20 categories (PAS-20). PASCAL-Context contains 5k validation images. We evaluate on the most frequent used 59 classes version (PC-59). Besides, we also annotate **Fantastic Beasts**, which contains 20 categories of human invented magical creatures from the Fantastic Beasts film series [1, 2, 3]. Detailed datasets' information can be found in the supplementary materials.

**Evaluation.** We report mean intersection-over-union (mIoU), following the recent open-vocabulary semantic segmentation (OVSS) literatures [25, 32, 11, 27, 10].

**Implementation Details.** We adopt CLIP ViT-L and ResNet101 as our backbone, and choose aggregation stages $L = 4$. Numbers of learnable cluster in each stage are $(15, 10, 5, 1)$. During training, the sampled attributes $N = 15$. AdamW optimizer is used with `CosineLRScheduler` by first warm up 10 epochs from initial learning rate $4e$-6 to $1e$-3, and the weight decay is set to 0.05.

## 4.1 Comparison with the State-of-the-art

Table 1: **Evaluation on PASCAL-**$5^i$ **and COCO-**$20^i$. For textual inputs, we compare "cls name" (category names) and "attr" (attributes). "direct", "pre-", "post-" and "hrchy" are four aggregation strategies in Sec. 3.4.4.

| Model | Settings | Backbone | PASCAL-$5^i$ | | | | | COCO-$20^i$ | | | | |
|---|---|---|---|---|---|---|---|---|---|---|---|---|
| | | | $5^0$ | $5^1$ | $5^2$ | $5^3$ | mIoU | $20^0$ | $20^1$ | $20^2$ | $20^3$ | mIoU |
| SPNet [46] | cls name | RN101 | 23.8 | 17.0 | 14.1 | 18.3 | 18.3 | - | - | - | - | - |
| ZS3Net [7] | cls name | RN101 | 40.8 | 39.4 | 39.3 | 33.6 | 38.3 | 18.8 | 20.1 | 24.8 | 20.5 | 21.1 |
| LSeg [25] | cls name | RN101 | 52.8 | 53.8 | 44.4 | 38.5 | 47.4 | 22.1 | 25.1 | 24.9 | 21.5 | 23.4 |
| LSeg [25] | cls name | ViT-L | 61.3 | 63.6 | 43.1 | 41.0 | 52.3 | 28.1 | 27.5 | 30.0 | 23.2 | 27.2 |
| LSeg [25] | attr (direct) | RN101 | 48.6 | 51.2 | 39.7 | 36.2 | 44.0 | 20.9 | 23.6 | 20.8 | 18.4 | 20.9 |
| LSeg [25] | attr (pre-) | RN101 | 49.1 | 51.4 | 40.9 | 35.9 | 44.3 | 21.7 | 24.5 | 22.0 | 19.6 | 22.0 |
| LSeg [25] | attr (post-) | RN101 | 50.0 | 52.2 | 42.1 | 36.8 | 45.3 | 21.2 | 24.0 | 22.5 | 19.2 | 21.7 |
| **AttrSeg (Ours)** | attr (hrchy) | RN101 | 52.9 | 55.3 | 45.0 | 43.1 | 49.1 | 27.6 | 28.4 | 26.1 | 22.7 | 26.2 |
| **AttrSeg (Ours)** | attr (hrchy) | ViT-L | **61.5** | **67.5** | **46.1** | **50.5** | **56.4** | **34.8** | **32.6** | **31.6** | **24.2** | **30.8** |

**PASCAL-**$5^i$ **and COCO-**$20^i$**.** We compare our method on PASCAL-$5^i$ and COCO-$20^i$ with state-of-the-art OVSS models SPNet [46], ZS3Net [7] and LSeg [25]. We also compare different textual input settings for LSeg. As shown in Tab. 1, our method outperform the competitive baselines. *Note that*, heavily relying on VLP that aligns image modality with its textual class name, it is undeniable that class names hold core and most direct information. Existing open-vocabulary segmentation models are inherently bound by this characteristic, *i.e.*, adding challenges when using only attributes that describe categories from indirect perspectives for segmentation. As a representative, LSeg shows weakness when the textual inputs are attributes descriptions only. In contrast, our method demonstrates good potential for segmenting the target object described by attributes through hierarchical aggregations, even surpass the SOTA with class name input.

**PASCAL-Context and PASCAL VOC.** We also evaluate our method on PASCAL-Context (PC-59) and PASCAL VOC (PAS-20), to compare with recent state-of-the-art OVSS works Zegformer [11], OVSeg [27] and CAT-Seg [10]. They are trained on a larger dataset COCO-Stuff [8] with more classes and more images. As shown in Tab. 2, our method still demonstrates superiority. Besides, compared with our method trained on COCO-Stuff and PASCAL VOC-15, the performance increased dramatically.

Table 2: **Evaluation on PASCAL-Context (PC-59) and PASCAL VOC (PAS-20) with attributes input using RN101 backbone**.

| Model | Training Data | PC-59 | PAS-20 |
|---|---|---|---|
| LSeg [25] | PASCAL VOC-15 | 24.2 | 44.0 |
| Fusioner [32] | PASCAL VOC-15 | 25.2 | 44.3 |
| **AttrSeg (Ours)** | PASCAL VOC-15 | **29.1** | **49.1** |
| Zegformer [11] | COCO-Stuff | 39.0 | 83.7 |
| OVSeg [27] | COCO-Stuff | 51.2 | 90.3 |
| CAT-Seg [10] | COCO-Stuff | 53.6 | 90.9 |
| **AttrSeg (Ours)** | COCO-Stuff | **56.3** | **91.6** |

Table 3: **Evaluation on Fantastic Beasts (using checkpoints transferred from PASCAL-$5^i$ and COCO-$20^i$).** For textual inputs, we compare "cls name" (category names) and "attr" (attributes). "direct", "pre-", "post-" and "hrchy" are four aggregation strategies. $5^i$ and $20^i$ refer to the best checkpoints from the $i^{th}$ fold for evaluation.

| Model | Settings | Backbone | PASCAL-$5^i$ | | | | | COCO-$20^i$ | | | | |
| --- | --- | --- | --- | --- | --- | --- | --- | --- | --- | --- | --- | --- |
| | | | $5^0$ | $5^1$ | $5^2$ | $5^3$ | mIoU | $20^0$ | $20^1$ | $20^2$ | $20^3$ | mIoU |
| LSeg [25] | cls name | RN101 | <10 | <10 | <10 | <10 | <10 | <10 | <10 | <10 | <10 | <10 |
| LSeg [25] | attr (direct) | RN101 | 44.8 | 46.6 | 46.4 | 46.2 | 46.0 | 50.1 | 51.2 | 51.7 | 49.3 | 50.6 |
| LSeg [25] | attr (pre-) | RN101 | 46.4 | 47.7 | 48.2 | 47.3 | 47.4 | 51.3 | 52.7 | 52.9 | 50.9 | 52.0 |
| LSeg [25] | attr (post-) | RN101 | 46.9 | 48.3 | 48.8 | 47.7 | 47.9 | 52.8 | 52.5 | 52.4 | 51.2 | 52.2 |
| **AttrSeg (Ours)** | attr (hrchy) | RN101 | 50.7 | 53.4 | 53.6 | 51.3 | 52.3 | 56.4 | 54.9 | 55.7 | 55.3 | 55.6 |
| **AttrSeg (Ours)** | attr (hrchy) | ViT-L | **54.1** | **55.8** | **55.4** | **54.5** | **55.0** | **59.2** | **58.8** | **58.3** | **58.5** | **58.7** |

**Fantastic Beasts.** We take Fantastic Beasts as an outstanding representative of the real-world that may encounter various situations like ambiguity, neologism and unnameability. To simulate real-world scenarios, we directly transfer the checkpoints of previous methods and ours trained from the their corresponding datasets for evaluation. As shown in Tab. 3 and Tab. 4, our method establish a new solid baseline. The existing SOTA methods are not able to handle these scenarios when taking class

Table 4: **Evaluation on Fantastic Beasts with attributes input using RN101 backbone (using checkpoints transferred from the corresponding training datasets).**

| Model | Training Data | mIoU (cls) | mIoU (attr) |
| --- | --- | --- | --- |
| LSeg [25] | PASCAL VOC-15 | <10 | 46.0 |
| Fusioner [32] | PASCAL VOC-15 | <10 | 46.1 |
| **AttrSeg (Ours)** | PASCAL VOC-15 | - | **52.3** |
| Zegformer [11] | COCO-Stuff | <20 | 55.7 |
| OVSeg [27] | COCO-Stuff | <20 | 58.1 |
| CAT-Seg [10] | COCO-Stuff | <20 | 59.4 |
| **AttrSeg (Ours)** | COCO-Stuff | - | **61.9** |

name as input. Despite using CLIP [42], which implicitly aligns visual and language features to some extent, these methods performance suffers due to the presence of unfamiliar words that break this alignment. However, when decomposing into attributes and then aggregating them, the performance of these methods remarkably increase, showing significant gains with robust performance. This demonstrates the universality and effectiveness of our motivation.

## 4.2 Ablation Study

**Various Aggregation Strategies.** Here, we present a comparison of different aggregation methods, as depicted in Tab. 5. The results demonstrate that: (1) "direct" gives the poorest results, suggesting that listing all attributes in one sentence cannot provide sufficient interactions between to-

Table 5: **Effectiveness of various aggregation strategies on PASCAL-$5^i$ with RN101 backbone.**

| Strat | LSeg | | | | | Ours | | | | |
| --- | --- | --- | --- | --- | --- | --- | --- | --- | --- | --- |
| | $5^0$ | $5^1$ | $5^2$ | $5^3$ | mIoU | $5^0$ | $5^1$ | $5^2$ | $5^3$ | mIoU |
| Direct | 48.6 | 51.2 | 39.7 | 36.2 | 44.0 | 48.8 | 51.4 | 39.7 | 39.2 | 44.8 |
| Pre- | 49.1 | 51.4 | 40.9 | 35.9 | 44.3 | 49.8 | 52.8 | 41.7 | 40.3 | 46.1 |
| Post- | 50.0 | 52.2 | 42.1 | 36.8 | 45.3 | 50.3 | 53.0 | 42.3 | 41.0 | 46.7 |
| Hrchy | - | - | - | - | - | **52.9** | **55.3** | **45.0** | **43.1** | **49.1** |

kens. (2) Both the "pre-" and "post-" both exhibit improvements compared with "direct". This may stem from the exchange of information between attribute and visual modalities within the open-vocabulary model. (3) In most cases, "Post-" performs better than "pre-", indicating that attribute aggregation after the model can facilitate greater interactions between different modalities. (4) It is not necessarily optimal to have an excessive or insufficient number of interactions throughout the entire process. Our proposed hierarchical aggregation method accounts for the regularity that the visual component learns differently at different stages. This necessitates its attribute counterpart to possess varying hierarchy. For instance, lower stages may focus on learning low-level features, requiring more attributes that represent local regions. Conversely, higher stages may concentrate on global information, necessitating fewer high-level attributes that are semantically abstract.

**Components in Aggregation Module.** We conduct ablation studies to investigate the importance of each component in our clustering module, as illustrated in Tab. 6. The result show that (1) the full module with all components achieves the best performance. (2) Cross attention on images contribute more compared with on attributes. As the aggregation among attributes finally aims to recognize the object in the image, it's important for the cluster tokens attending to the visual

Table 6: **Ablation of components in clustering module on PASCAL-$5^i$ with ViT-L backbone.**

| Comp | Cross Attn | | Mixer | $5^0$ | $5^1$ | $5^2$ | $5^3$ | mIoU |
| --- | --- | --- | --- | --- | --- | --- | --- | --- |
| | Img | Attr | | | | | | |
| Full | ✓ | ✓ | ✓ | **61.5** | **67.5** | **46.1** | **50.5** | **56.4** |
| Cross Attn | ✗ | ✓ | ✓ | 59.2 | 65.6 | 44.1 | 47.4 | 54.1 |
| | ✓ | ✗ | ✓ | 59.6 | 65.8 | 44.4 | 48.6 | 54.6 |
| | ✗ | ✗ | ✓ | 58.6 | 65.0 | 42.9 | 46.4 | 53.2 |
| Mixer | ✓ | ✓ | ✗ | 60.7 | 67.0 | 45.6 | 49.6 | 55.7 |
| Mini | ✗ | ✗ | ✗ | 58.4 | 64.9 | 42.4 | 46.0 | 52.9 |

feature. (3) If no cross attentions given, the performance further degraded. This demonstrates the

cross attentions on two modalities enable the learnable cluster centers match better with the correct attributes. (4) Mixer also plays a role in the bottom part of the module, as it helps to propagate and exchange information between clusters for further aggregation. (5) When neither of these is applied, only the SlotAttn is introduced, which, unsurprisingly, yields the poorest result.

**Numbers of Decomposed Attributes.** Tab. 7 shows the results when given input numbers of attribute (#Attr) changes. As #Attr decreases, the available information diminishes, potentially resulting in an incomplete object description. In general, such a reduction in attribute quantity can lead to a significant decline in performance. However, our method demonstrates a certain level of robustness even when #Attr decreases from 15 to 10, and even 5. This resilience can be ascribed to the

Table 7: **Ablation of #Attributes inputs and #Stages on PASCAL-$5^i$ with ViT-L backbone**.

| #Attr | #Stage | #Tokens/Stage | $5^0$ | $5^1$ | $5^2$ | $5^3$ | mIoU |
|---|---|---|---|---|---|---|---|
| 15 | 5 | $(15, 10, 5, 3, 1)$ | 61.0 | 67.3 | **46.9** | **50.7** | **56.5** |
| | 4 | $(15, 10, 5, 1)$ | **61.5** | **67.5** | 46.1 | 50.5 | 56.4 |
| | 3 | $(15, 10, 1)$ | 60.1 | 66.2 | 44.7 | 48.9 | 55.0 |
| | 2 | $(15, 1)$ | 57.8 | 63.5 | 41.9 | 46.4 | 52.4 |
| 10 | 5 | $(10, 5, 3, 2, 1)$ | 59.4 | 65.8 | **44.9** | 49.2 | 54.8 |
| | 4 | $(10, 5, 3, 1)$ | **59.6** | **66.1** | 44.7 | **49.9** | **55.1** |
| | 3 | $(10, 5, 1)$ | 58.0 | 64.8 | 43.8 | 47.1 | 53.4 |
| | 2 | $(10, 1)$ | 55.9 | 62.8 | 42.5 | 45.4 | 51.7 |
| 5 | 5 | $(5, 4, 3, 2, 1)$ | 51.8 | 62.9 | 41.9 | 44.8 | 50.4 |
| | 4 | $(5, 3, 2, 1)$ | **52.4** | **63.1** | **43.7** | 45.2 | **51.1** |
| | 3 | $(5, 3, 1)$ | 52.1 | 62.7 | 43.0 | **45.4** | 50.8 |
| | 2 | $(5, 1)$ | 49.2 | 61.6 | 42.1 | 44.2 | 49.3 |

fact that the attributes are randomly sampled (with repetition) from the attribute set during training. Consequently, the attributes describing a specific category do not necessarily need to be complete or highly informative for the model to generate accurate outputs.

**Numbers of Aggregation Stages.** As shown in Tab. 7, we reduce the numbers of stage (#Stage) from 5 to 2. In each stage, we attempt to aggregate to half of the tokens until there is only one left. In most cases, less aggregation stages cause deficient results, and more stages result in better performance. However, blindly increasing stages do not necessarily lead to better performance. For #Attr is 5 or 10, 5 stages aggregation lead to a decrease compared with 4 stages. A reason may be that too much aggregation operation is excessive for limited attributes, which may disrupt the hierarchy inside, thus negatively affects the result. Considering the trade-off, we choose 4 stages in our method.

**Impact of Inaccurate/Incorrect Attributes.** In real-world scenarios, attribute decomposition may include slight noise. Our aggregation module exhibits a certain level of robustness to noise during both training and inference. (1) For *inaccurate* attributes (attributes that are not visible in a specific image, but are still related to the class, such as "four-legged" to "dog" in a dog lying down image), during training, we randomly select attributes from the attribute pool for a given class, which means that the model is trained with potentially noisy and inaccurate at-

Table 8: **Ablation of the impact of inaccurate/incorrect attributes during decomposition, and the VLM filtering strategy**.

| Clean Attr | Inaccurate Attr | Incorrect Attr | VLM Filtering | mIoU |
|---|---|---|---|---|
| ✓ | | | | 59.5 |
| ✓ | ✓ | | | 59.1 |
| ✓ | ✓ | ✓ | | 55.4 |
| ✓ | ✓ | ✓ | ✓ | 58.9 |

tributes. However, as demonstrated in Tab. 8, our model can learn to ignore these inaccurate attributes during aggregation, and instead focus on other attributes to produce correct segmentation results. (2) For *incorrect* attributes (attributes that are completely unrelated to the target class, such as "red" to "dog"), a naive approach would be to first filter the input attributes using existing VLMs, and then select the top related attributes for downstream processing. Tab. 8 assess the impact of incorrect attributes and VLM filtering on inference. The results indicate a simple VLM filtering has the effect.

**Types of Decomposed Attributes.** We roughly categorize the decomposed attributes into four types: color, shape, parts, and others, and maintain a consistent total number of inputs. As shown in Tab. 9, all types of attributes contribute to the overall performance, highlighting the significance of attribute diversity.

Table 9: **Ablation of decomposed attributes' types on Fantastic Beast with RN101 backbone**.

| Color | Shape | Parts | Others | mIoU |
|---|---|---|---|---|
| ✓ | | | | 30.8 |
| ✓ | ✓ | | | 42.5 |
| ✓ | ✓ | ✓ | | 51.0 |
| ✓ | ✓ | ✓ | ✓ | 52.3 |

## 5 Conclusion

We pioneer the early exploration in utilizing only attribute descriptions for open-vocabulary segmentation, and provide detailed attribute descriptions for two types of existing datasets and one newly collected dataset. Based on this, we propose a novel attribute decomposition-aggregation framework that first decouples class names into attribute descriptions, and then combines them into final class representations. Extensive experiments demonstrate the effectiveness of our method, showcasing its ability to achieve the state-of-the-art performance across various scenarios.

# 6 Acknowledgement

This work is supported by the National Key R&D Program of China (No. 2022ZD0160703), STCSM (No. 22511106101, No. 22511105700, No. 21DZ1100100), 111 plan (No. BP0719010) and National Natural Science Foundation of China (No. 62306178).

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
