# Supplementary Materials:
# AttrSeg: Open-Vocabulary Semantic Segmentation via Attribute Decomposition-Aggregation

**Chaofan Ma**[1], **Yuhuan Yang**[1], **Chen Ju**[1], **Fei Zhang**[1], **Ya Zhang**[1,2], **Yanfeng Wang**[1,2✉]

[1] Coop. Medianet Innovation Center, Shanghai Jiao Tong University
[2] Shanghai AI Laboratory

{chaofanma, yangyuhuan, ju_chen, ferenas, ya_zhang, wangyanfeng622}@sjtu.edu.cn

## 1 PASCAL-5$^i$ and COCO-20$^i$ with Attributes

### 1.1 Motivations

As we pioneer the exploration of segmentation through attributes, there is currently no existing benchmark that fulfills our requirements. To address this, we annotate attributes on widely used datasets such as PASCAL [4, 5] and COCO [7].

For training, these datasets provide representative simulation environments. They consist of image-mask-attribute pairs and are organized into well-defined folds for cross-validation. Models trained on this dataset demonstrate successful adaptation to real-world scenarios, including cases involving neologisms or unnameability. This substantiates the rationality and practicality of this environment.

For evaluation, PASCAL and COCO serve as standard benchmarks to showcase the effectiveness of our approach. Additionally, these datasets facilitate the comparison and assessment of performance against other established baselines, thereby enhancing our understanding of the field. Moreover, the attribute annotations for these datasets can simulate scenarios where vanilla category names are semantically coarse or ambiguous.

### 1.2 Category Names

In Tab. 1, we provide the detailed categories split settings used in our experiments. Datasets PASCAL-5$^i$ [4, 5] and COCO-20$^i$ [7] follow the split settings proposed in LSeg [6].

Table 1: **Categories split for PASCAL-5$^i$ and COCO-20$^i$**. Categories in each fold are for test, and classes in the rest three folds are used for training.

| Dataset | fold 0 | fold 1 | fold 2 | fold 3 |
|---|---|---|---|---|
| PASCAL-5$^i$ | aeroplane, bicycle, bird, boat, bottle | bus, car, cat, chair, cow | diningtable, dog, horse, motorbike, person | pottedplant, sheep, sofa, train, tvmonitor |
| COCO-20$^i$ | person, aeroplane, boat, parkingmeter, dog, elephant, backpack, suitcase, sportsball, skateboard, wineglass, spoon, sandwich, hotdog, chair, diningtable, mouse, microwave, refrigerator, scissors | bicycle, bus, trafficlight, bench, horse, bear, umbrella, frisbee, kite, surfboard, cup, bowl, orange, pizza, sofa, toilet, remote, oven, book, teddybear | car, train, firehydrant, bird, sheep, zebra, handbag, skis, baseballbat, tennisracket, fork, banana, broccoli, donut, pottedplant, tvmonitor, keyboard, toaster, clock, hairdrier | motorbike, truck, stopsign, cat, cow, giraffe, tie, snowboard, baseballglove, bottle, knife, apple, carrot, cake, bed, laptop, cellphone, sink, vase, toothbrush |

37th Conference on Neural Information Processing Systems (NeurIPS 2023).

### 1.3 Questions Templates and Prompts

To automatically adapt large language models (LLMs) as attribute description generators, below are several question templates asking LLMs:

```
T1: Describe what a {category} looks like in the image.
T2: How can you identify a {category} in the image?
T3: What are the characteristics of a {category} in the image?
T4: What are visual features of a {category} in the image?
T5: List all attributes for distinguishing a {category} in a photo.
```

We can combine the answers for all these templates. However, the output of these questions is in a random format, so it is still needed to manually parse into clean attribute. To enable the LLMs to produce attribute descriptions in a way that does not require so much human parsing, we choose to give the desired response as the prompt. Here is an example for using the first question template (T1):

```
Q: Describe what a flamingo looks like in the image.
A: wading bird; pink or reddish color; long legs; long neck; curved
    beak; webbed feet; black-tipped wings; black flight feathers; pink
     or red eyes.

Q: Describe what a {category} looks like in the image. Give me the
    answer in the above pattern.
A:
```

where `{category}` is substituted for a specific class in the dataset, as mentioned in Sec. 1.2. In this way, we can parse the answer automatically by split from semicolons.

### 1.4 Attributes Examples

We here provide some attributes of PASCAL as an example.

```
aeroplane: ["metal body", "wings", "tail", "propellers or jet engines",
     "propellers", "jet engines", "landing gear", "cockpit", "windows",
     "fuselage", "control surfaces", "navigation lights"]

bicycle: ["two-wheeled", "metal frame", "handlebars", "pedals", "seat",
     "chain", "wheels", "brakes", "reflectors", "gears", "lights", "
    kickstand"]

bird: ["two wings", "two legs", "a beak", "feathers", "a tail", "
    bright colors", "a distinctive call or song", "a curved neck", "
    small eyes", "a pointed head", "a small body", "a pointed beak", "
    webbed feet"]
```

## 2 Fanatic Beasts Datasets

### 2.1 Motivations

This dataset is collected with the intention of conducting a comprehensive evaluation and simulating real-world scenarios.

Existing datasets typically lack the inclusion of rare or obscure vocabulary. To address this limitation, we manually curate a dataset comprising human-made objects and rare categories. This dataset is specifically designed to simulate two common scenarios where attribute descriptions are necessary:

**Neologisms.** Vanilla category names are new vocabularies, which are unseen by large language models (LLMs) and vision-language pre-trainings (VLPs).

**Unnameability.** Users are not familiar with an object, they may have difficulty naming it, especially when it comes to a rare or obscure category.

Table 2: **Performance of each category in Fantastic Beasts on CLIP.**

| Categories | Augurey | Billywig | Chupacabra | Diricawl | Doxy | Erumpent | Fwooper | Graphorn | Grindylow | Kappa | Leucrotta | Matagot | Mooncalf | Murtlap | Nundu | Occamy | Runespoor | Swoopingevil | Thunderbird | Zouwu | Average |
|---|---|---|---|---|---|---|---|---|---|---|---|---|---|---|---|---|---|---|---|---|---|
| Acc | 0.20 | 0.11 | 0.22 | 0.17 | 0.17 | 0.09 | 0.10 | 0.07 | 0.15 | 0.08 | 0.00 | 0.00 | 0.07 | 0.00 | 0.00 | 0.00 | 0.16 | 0.17 | 0.25 | 0.10 | **0.11** |

## 2.2 Details

**Category Names and Attributes.** We collect total 27 categories of magical creatures from the film series of Fantastic Beasts, and choose 20 of them which are the least recognizable (the top 20 rarest), listed as below in alphabetical order:

> Augurey, Billywig, Chupacabra, Diricawl, Doxy, Erumpent, Fwooper, Graphorn, Grindylow, Kappa, Leucrotta, Matagot, Mooncalf, Murtlap, Nundu, Occamy, Runespoor, Swoopingevil, Thunderbird, Zouwu

Attributes for each category are carefully labelled by fan volunteers of the film series, most of which have greater than 10 attributes.

**Images and Masks.** For each category, most of the categories have greater than 10 images. Most of the images are collected from the web, and some are screenshot from the films. Masks are also carefully annotated by fan volunteers of the film series, and most of the images have one corresponding mask per image. Since all these creatures and names are human inventions, they are unlikely to been learned by existing LLMs and VLPs.

## 2.3 Recognizability by VLPs

To demonstrate that the VLPs are unfamiliar with the categories in our dataset, we employ CLIP [8] to classify these categories. As presented in Tab. 2, the average accuracy is only 0.11, which is significantly lower than the accuracy on other datasets reported. These findings indicate that the categories in this dataset are rare and obscure, and are not familiar to the VLPs.

## 2.4 Brief Introduction of Fantastic Beasts Film Series

The Fantastic Beasts film series is a spin-off prequel to the Harry Potter novel and film series. The series is directed by David Yates and distributed by Warner Bros. It consists of three fantasy films as of 2022:

**Fantastic Beasts and Where to Find Them (2016) [1].** The movie follows Newt Scamander, a magizoologist who travels to New York with a suitcase full of magical creatures. When some of the creatures escape, he teams up with a group of people to find them before they cause any harm.

**Fantastic Beasts: The Crimes of Grindelwald (2018) [2].** The movie follows Newt Scamander as he teams up with Albus Dumbledore to stop the dark wizard Gellert Grindelwald from raising an army of pure-blood wizards to rule over non-magical beings.

**Fantastic Beasts: The Secrets of Dumbledore (2022) [3].** The movie follows Newt Scamander as he teams up with Albus Dumbledore once again to stop the dark wizard Gellert Grindelwald from obtaining a powerful object that could destroy the world.

This movie series is a good choice for our experiments because it contains many human-invented fantastic beasts with different appearances and category names.

## 2.5 Copyright

Once this dataset is released, it will possess non-commercial licenses that are granted free-of-charge to qualified researchers, academics, and non-commercial developers for non-profit research or development purposes.

According to Digital Media Law Project and Legal Information Institute (Section 107 of the Copyright Act), the "fair use" of a copyrighted work, including such use by reproduction in copies or phonorecords or by any other means specified by that section, for purposes such as criticism, comment, news reporting, teaching (including multiple copies for classroom use), scholarship, or research, is not an infringement of copyright. We believe the collection and usage of this dataset should be considered as "fair use".

## 3 Visualizations

We here show visualizations of images, predicted segmentation masks, category names, and some corresponding main attributes on Fantastic Beasts dataset, as presented in Fig. 1, Fig. 2, Fig. 3 and Fig. 4. It should be noted that our model is ***not*** trained on this dataset, and the shown results were evaluated using the COCO-20$^i$ checkpoints.

## 4 Limitations and Broader Impacts

Our model utilizes CLIP, which has been trained on images and text data collected from the Internet. As a result, the model predicts content based on the learned statistics of the training dataset, which may reflect biases present within the data, including those with negative societal impacts. Consequently, the model may generate inappropriate and inaccurate results.

Moreover, our approach employs large language models (LLMs) to generate attribute descriptions, which may potentially introduce biases if not properly regulated.

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

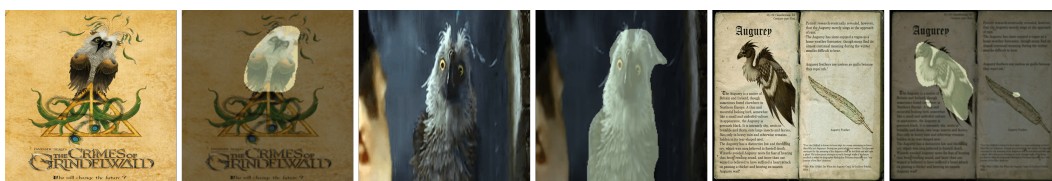

(a) **Category**: Augurey. **Main Attributes**: "greenish-black feathers", "long, sharp beak", "bird-like creature", "yellow eyes", "resembles a thin and underfed vulture"

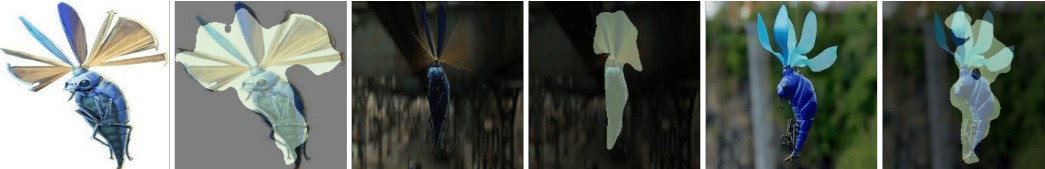

(b) **Category**: Billywig. **Main Attributes**: "blue or green or black color", "a small, insect-like body", "a stinger on its tail", "long antennae", "spiny projections along its sides", "long, thin wings that spin rapidly"

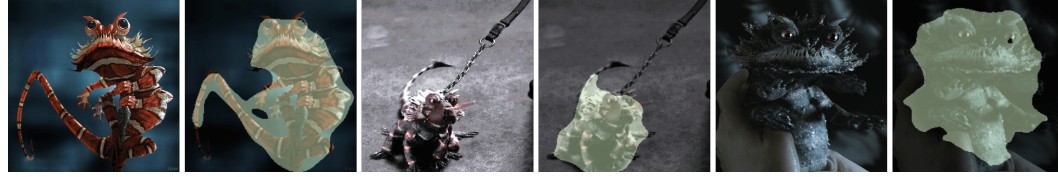

(c) **Category**: Chupacabra. **Main Attributes**: "part-lizard", "part-homunculus", "blue or red skin", "six legs", "multiple spines", "several sharp teeth", "long tail", "round eyes", "blue markings with red rings on its body"

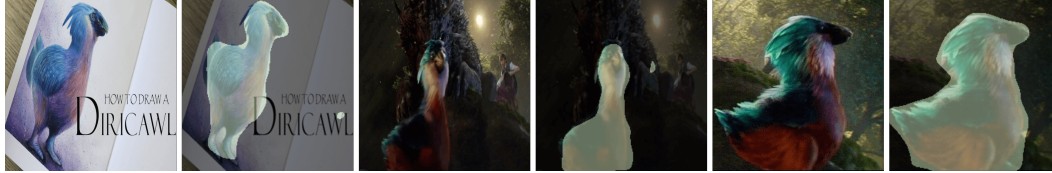

(d) **Category**: Diricawl. **Main Attributes**: "light blue and light pink feathers", "small, plump, flightless bird", "brown in color", "white feathers on its breast", "fluffy-feathered bird"

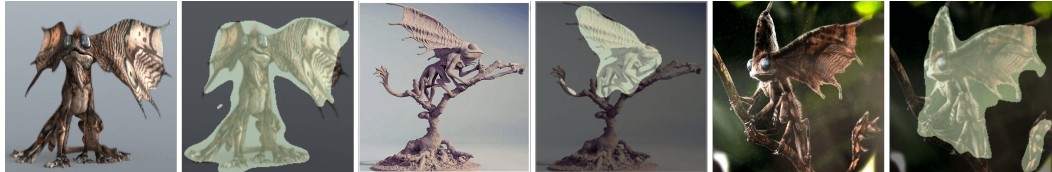

(e) **Category**: Doxy. **Main Attributes**: "small, fairy-like creature", "thick, black hair", "four beetle-like wings", "a pair of sharp pincers", "four arms and four legs", "two rows of teeth", "round eyes"

Figure 1: **Visualizations of Fantastic Beasts (Part 1/4).** Images, predicted segmentation masks, category names, and some corresponding main attributes are presented.

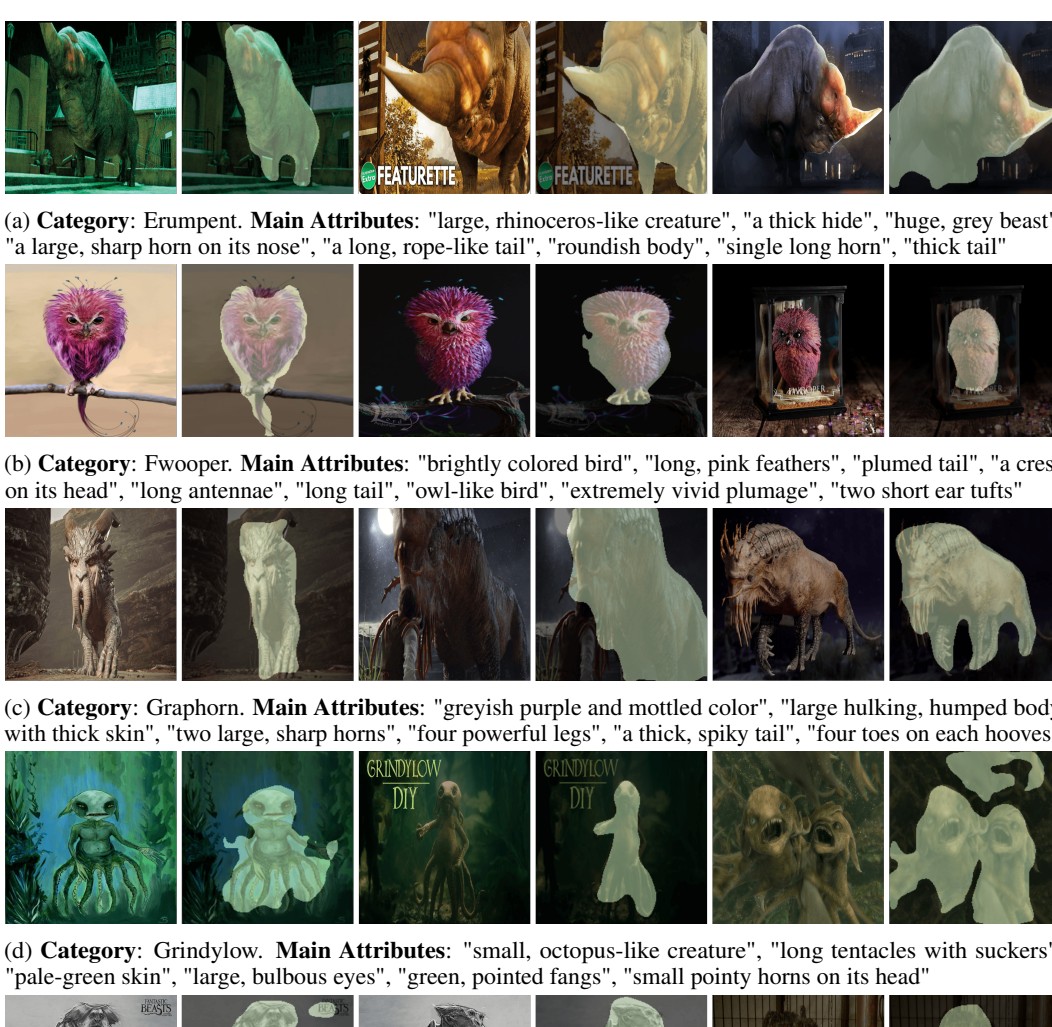

(a) **Category**: Erumpent. **Main Attributes**: "large, rhinoceros-like creature", "a thick hide", "huge, grey beast", "a large, sharp horn on its nose", "a long, rope-like tail", "roundish body", "single long horn", "thick tail"

(b) **Category**: Fwooper. **Main Attributes**: "brightly colored bird", "long, pink feathers", "plumed tail", "a crest on its head", "long antennae", "long tail", "owl-like bird", "extremely vivid plumage", "two short ear tufts"

(c) **Category**: Graphorn. **Main Attributes**: "greyish purple and mottled color", "large hulking, humped body with thick skin", "two large, sharp horns", "four powerful legs", "a thick, spiky tail", "four toes on each hooves"

(d) **Category**: Grindylow. **Main Attributes**: "small, octopus-like creature", "long tentacles with suckers", "pale-green skin", "large, bulbous eyes", "green, pointed fangs", "small pointy horns on its head"

(e) **Category**: Kappa. **Main Attributes**: "like a humanoid turtle", "water-dwelling creature", "scaly, green skin", "sharp claws and beak", "scaly monkey with webbed hands and feet", "a water-filled hollow on top of its head"

Figure 2: **Visualizations of Fantastic Beasts (Part 2/4).** Images, predicted segmentation masks, category names, and some corresponding main attributes are presented.

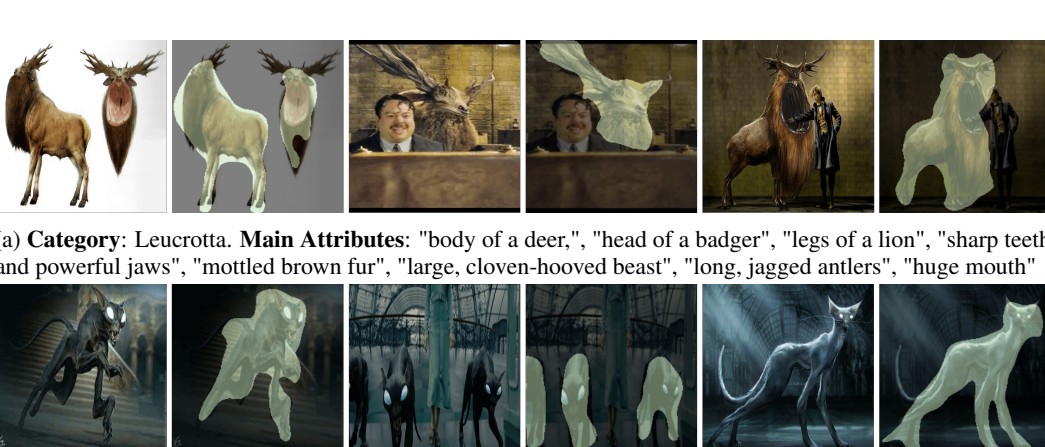

(a) **Category**: Leucrotta. **Main Attributes**: "body of a deer,", "head of a badger", "legs of a lion", "sharp teeth and powerful jaws", "mottled brown fur", "large, cloven-hooved beast", "long, jagged antlers", "huge mouth"

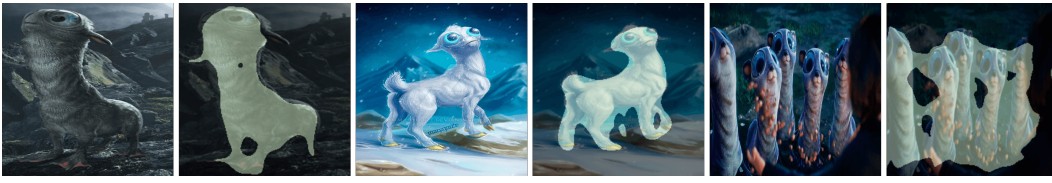

(b) **Category**: Matagot. **Main Attributes**: "glowing yellow eyes", "a cat-like creature", "black fur", "big, blue eyes", "sharp teeth", "large hairless black cat-like creatures"

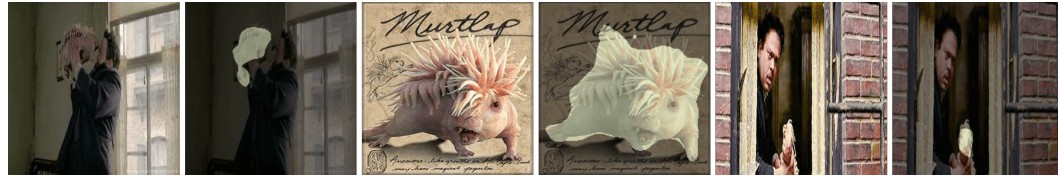

(c) **Category**: Mooncalf. **Main Attributes**: "large, bulging, expressive eyes", "long, spindly legs", "a round head", "smooth, pale-grey skin", "narrow legs with large hooves", "long necks", "large flat feet"

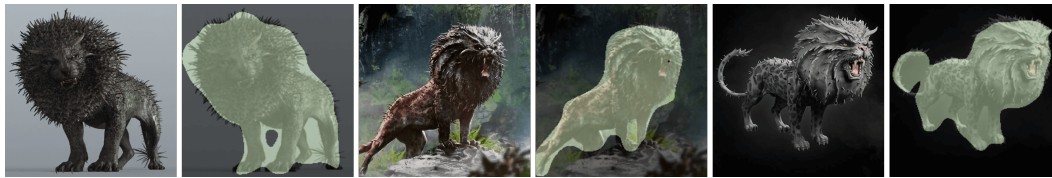

(d) **Category**: Murtlap. **Main Attributes**: "small, rodent-like creature", "a tough, rubbery hide", "long, sharp claws", "a row of spines like a sea anemone along its back", "pale pink skin", "pink or red eyes", "sharp teeth"

(e) **Category**: Nundu. **Main Attributes**: "large, ferocious feline creature", "golden fur with black spots", "large, sharp teeth", "resembles a leopard", "thick mane", "spiky fur"

Figure 3: **Visualizations of Fantastic Beasts (Part 3/4).** Images, predicted segmentation masks, category names, and some corresponding main attributes are presented.

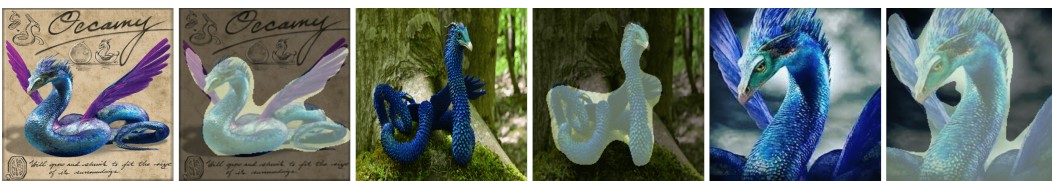

(a) **Category**: Occamy. **Main Attributes**: "a feathered, bird-like head and wings", "serpentine body", "turquoise skin", "purple feathers", "a plumed, two-legged winged creature", "long, thin beak"

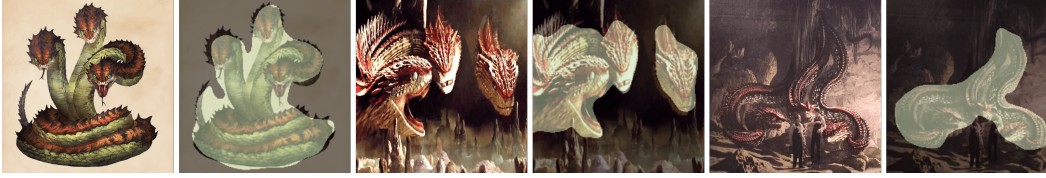

(b) **Category**: Runespoor. **Main Attributes**: "serpent-like creature", "three differently sized heads", "orange, green, and purple color on each head", "long, thin body with scales", "forked tongue", "orange and black stripes"

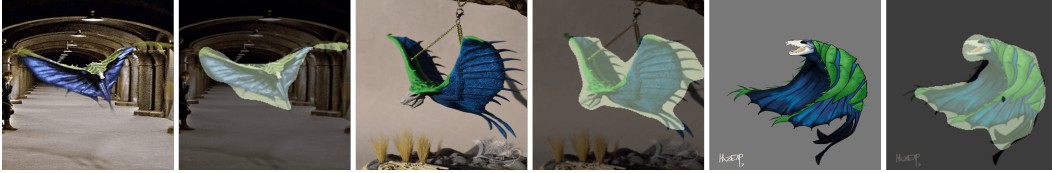

(c) **Category**: Swoopingevil. **Main Attributes**: "large butterfly-like creature", "head like a wolf's skull", "blue and green wings", "a long, thin body", "a sharp beak", "bat-like wings that can expand like a huge butterfly"

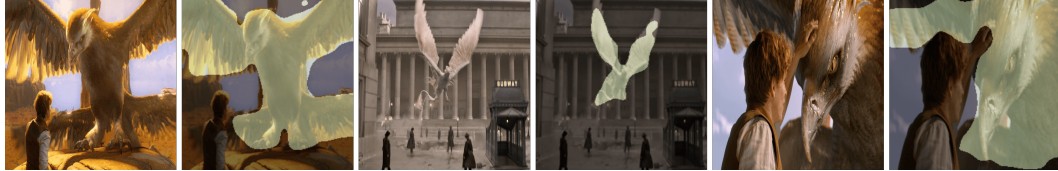

(d) **Category**: Thunderbird. **Main Attributes**: "large, bird-like creature", "white feathers with gold patterns", "sharp eyes", "head similar to an eagle", "possess three pairs of powerful wings", "a long, thin beak"

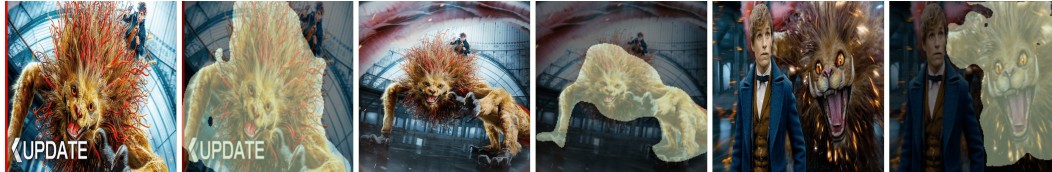

(e) **Category**: Zouwu. **Main Attributes**: "large, feline-like creature", "a striped, multicolored body", "a long, serpentine tail", "head of a tiger", "scraggly mane", "fangs extend and curl out of its mouth", "long sharp claws"

Figure 4: **Visualizations of Fantastic Beasts (Part 4/4).** Images, predicted segmentation masks, category names, and some corresponding main attributes are presented.