# OpenReview forum: "AttrSeg: Open-Vocabulary Semantic Segmentation via Attribute Decomposition-Aggregation"
_NeurIPS.cc/2023/Conference — NeurIPS 2023 poster_

### Official Review · Reviewer_PB8W · 2023-07-01

**Soundness:** 2 fair
**Presentation:** 2 fair
**Contribution:** 2 fair
**Rating:** 5
**Confidence:** 4

**Summary:**

The paper proposes a decomposition-aggregation framework to address the problem of open-vocabulary semantic segmentation. The framework consists of a decomposition stage, where pre-trained language models (LLMs) such as ChatGPT are used to extract textual attributes for class names, and an aggregation stage, where a hierarchical structure is employed to progressively aggregate attribute features. The experimental results demonstrate the effectiveness of the proposed method.

**Strengths:**

1. The idea behind the framework is straightforward and easy to understand.
2. The experimental results provide evidence that the proposed method is effective.

**Weaknesses:**

1. The performance of the framework heavily relies on the quality and accuracy of the selected attributes. Incorrect or inaccurate attributes may degrade the segmentation performance. It would be beneficial to explore ways to regulate the risk introduced during the decomposition stage, both during training and inference.
2. While leveraging additional attributes from external knowledge improves the performance of open-vocabulary semantic segmentation (OVSS), it is important to consider whether explicitly decomposing class names into attributes align with the long-term goals of OVSS. Implicitly learning the "decomposition and composition" process might be a more desirable solution in the future.

**Questions:**

When decomposing a class name into a set of attributes using a language model like ChatGPT, is it correct to say that the attributes generated by the language model are based on common sense rather than visual inspection, considering that the language model does not have access to the images?

**Limitations:**

Refer to Weaknesses

---

> ### Author Rebuttal · Authors · 2023-08-10
>
>
> ## Reviewer PB8W
>
> Thank you for your valuable comments and suggestions, as well as acknowledging the novelty and results of the idea.
> We address individual questions and concerns below.
>
>
>
> > **1. It would be beneficial to explore ways to regulate the risk introduced during the decomposition stage, both during training and inference.**
>
>
> We agree that in real-world scenarios, attribute decomposition may include slight noise.
> We considered this when designed, so our aggregation module exhibits a certain level of robustness to noise during both training and inference.
>
> (1) For inaccurate attributes (attributes that are not visible in a specific image, but are still related to the class, such as "four-legged" to "dog" in a dog lying down image):
> During training, we randomly select attributes from the attribute pool for a given class, which means that the model is trained with potentially noisy and inaccurate attributes.
> However, as demonstrated in Table 7 below, our model can learn to ignore these inaccurate attributes during aggregation, and instead focus on other attributes to produce correct segmentation results.
>
> (2) For incorrect attributes (attributes that are completely unrelated to the target class, such as "red" to "dog"):
> A naive approach would be to first filter the input attributes using existing VLP models such as CLIP, and then select the top related attributes for downstream processing.
> We conducted an ablation study to assess the impact of incorrect attributes and VLM filtering on inference.
> As shown in the table 7 below, the results indicate a simple VLM filtering has the effect.
>
> More in-depth investigations belong to the area of "learning with noisy labels", which is beyond the scope of this paper.
> We leave it as future work.
>
>
> *Table 7: Ablation of the impact of inaccurate/incorrect attributes and VLM filtering during decomposition.*
> | Clean attr | Inaccurate attr | Incorrect attr | VLM filtering | mIoU |
> | :--------: | :-------------: | :------------: | :-----------: | :--: |
> |     √      |                 |                |               | 59.5 |
> |     √      |        √        |                |               | 59.1 |
> |     √      |        √        |       √        |               | 55.4 |
> |     √      |        √        |       √        |       √       | 58.9 |
>
>
>
>
>
> > **2. Implicitly learning the "decomposition and composition"**
>
>
> We appreciate the reviewer's suggestion regarding the potential future solution of implicitly learning decomposition and composition.
>
> Since the specific method for implicitly learning the "decomposition and composition" has not been explicitly mentioned by the reviewer,
> we propose one possible approach for implicit modeling.
> In this approach, given a category name as input, the model outputs N category representations, with each representation representing an attribute.
> These attributes are then constrained by a loss function to ensure orthogonality among them.
>
> However, we believe that learning implicit attributes from raw data is not a trivial task and faces several limitations.
> It incurs high computational costs, lacks interpretability, lacks direct constraints, introduces potential noise, and is hard to evaluating the quality and relevance of implicit attributes.
>
> These challenges can be addressed more easily by explicitly decomposing attributes.
> Our work, as a pioneer in this area, mainly focuses on demonstrating the feasibility and effectiveness of attribute decomposition-aggregation.
> We believe that our method can serve as a solid baseline approach in this field.
> Besides, we also provide a novel perspective for the community to address the existing limitations, thus making a significant step towards the ultimate goal of OVSS.
>
> Future research could explore how to move beyond explicit attributes and instead learn representations that implicitly capture the compositional structure of novel object categories.
>
>
>
>
> > **3. is it correct to say that the attributes generated by the language model are based on common sense rather than visual inspection?**
>
> Yes, this is correct.
>
> LLMs can successfully imitate visual attributes since they are trained on a large corpus containing descriptions with visual knowledge.
> Besides in LLM, a high-dimensional knowledge graph potentially exists, enabling the decomposition of text into attributes that do not necessarily depend on visual elements.
>
> When multi-modal foundation models, such as multi-modal GPT-4, become more advanced, the attribute generation may be able to base on common sense together with image content.

---

> > ### Comment · Reviewer_PB8W · 2023-08-12
> >
> > I thank the authors for the detailed response. I have also read through all reviews given by other reviewers and the author's corresponding responses. The paper looks more interesting and complete if all the additional experimental results are added. Even though the method for this work is straightforward, I do believe that the value and inspiration this work will bring to the community.

---

> > > ### Author Response · Authors · 2023-08-17
> > >
> > > Thanks for the support to our paper!
> > >
> > > All the additional experimental results will definitely be added in our revised paper, and thank you again for considering our responses.

---

### Official Review · Reviewer_mZMs · 2023-07-07

**Soundness:** 3 good
**Presentation:** 3 good
**Contribution:** 3 good
**Rating:** 6
**Confidence:** 4

**Summary:**

[Task] This paper presents a novel decomposition-aggregation framework for open-vocabulary semantic segmentation, which addresses the challenges of segmenting novel object categories during inference. The existing vision-language pre-training approaches make unrealistic assumptions about low-quality textual category names, such as the availability of accurate and complete category descriptions in pre-trained lexicons. This work aims to overcome these limitations and proposes a framework inspired by human cognition for understanding new concepts.

[Method] The framework consists of two stages: decomposition and aggregation. In the decomposition stage, class names are decoupled into diverse attribute descriptions to enrich semantic contexts. Two attribute construction strategies are employed: leveraging large language models for common categories and manual labeling for human-invented categories. In the aggregation stage, diverse attributes are grouped into an integrated global description, forming a discriminative classifier that distinguishes the target object from others. A hierarchical aggregation technique is introduced to achieve multi-level alignment and deep fusion between vision and text. The final segmentation result is obtained by computing the embedding similarity between aggregated attributes and images.

[Experiments] To evaluate the effectiveness of the proposed approach, three datasets with attribute descriptions are annotated, and extensive experiments and ablation studies are conducted. The results demonstrate the superior performance of the attribute decomposition-aggregation framework. The paper concludes by mentioning the release of datasets and codes upon publication, which will enable further exploration and validation of the proposed method.

**Strengths:**

[Novel Framework]: The paper introduces a novel decomposition-aggregation framework that addresses the challenges of open-vocabulary semantic segmentation. By drawing inspiration from cognitive psychology, the framework leverages diverse attribute descriptions to enhance category comprehension and improve segmentation performance.

[Practical Relevance]: The paper highlights the limitations of existing vision-language pre-training approaches in real-world scenarios and addresses issues such as ambiguity, neologisms, and unnameability, which are critical for achieving accurate and comprehensive open-vocabulary semantic segmentation results.

[Construction Strategies]: The paper proposes two construction strategies for generating attribute descriptions, utilizing language models and manual labeling. This provides flexibility and accommodates different scenarios, including both common categories and situations involving novel or difficult-to-describe categories. The inclusion of a hierarchical aggregation architecture enables the exploitation of potential attribute hierarchy, which can enhance the segmentation process and improve the quality of the final segmentation result.

[Experimental Evaluation]: The paper evaluates the proposed framework on multiple datasets, demonstrating its superior performance over various baselines and competitors. The inclusion of thorough ablation studies further enhances the understanding of each component's contribution.

**Weaknesses:**

[Weak baselines] The paper primarily compares the proposed work to LSeg, which may be considered a weak baseline as it is not the latest work in the field of open-vocabulary semantic segmentation. The achieved performance gains with the proposed framework, using RN101 as a backbone, are relatively modest at around 3% (3% is modest since the baseline is LSeg rather than the current SOTA work in open-vocab semantic segmentation). Given the complexity of the framework and training pipeline, it raises questions about the overall effectiveness of the proposed approach. Can you compare your work to some latest works as well?

[Performance evaluation with class names alone] The experimental results presented in Tables 2 and 3 demonstrate that the performance on the custom datasets Pascal-5 and COCO-20 is only comparable to LSeg (LSeg is evaluated based on class names). While the capability of evaluating the model's performance without relying on class names is intriguing, it is more common and practically useful to support evaluation with class names. Therefore, could you please clarify whether the proposed approach's performance would be influenced when class names are provided as part of the inputs?

**Questions:**

My main questions are listed in the weakness sections. I have a few more questions on the ablation study and model evaluation as follows:

In addition to evaluating with exact class names, it would be beneficial to know if the proposed approach can handle hybrid inputs. For instance, can it work with rough class names such as "dog" without specifying the breed (e.g., "corgi"), combined with additional textual descriptions? Supporting hybrid inputs would enhance the flexibility and practicality of the proposed framework, enabling more realistic scenarios where users might provide partial or general class names along with descriptions. Addressing these concerns would provide a better understanding of the comparative performance of the proposed framework and its compatibility with evaluating segmentation results using hybrid inputs.

**Limitations:**

Yes, the authors adequately addressed the limitations.

---

> ### Author Rebuttal · Authors · 2023-08-10
>
> ## Reviewer mZMs
>
> Thank you for your appreciations on the novelty, contribution, and results of the idea, as well as the detailed and constructive review.
> We address individual questions and concerns below.
>
>
> > **1. compare with some latest works**
>
>
> Thanks for the suggestions, we have added the comparison with the latest works including Fusioner, Zegformer, OVSeg, and CAT-Seg as also suggested by R-cgtT and R-fEW5. This is shown in Table 1, 2 in the general response.
>
> We would like to classify the reason of our method seems relatively modest in improvement.
> Most of latest works are trained on a larger dataset COCO-Stuff with more classes and more images (118k densely annotated training images with 171 categories), while ours are strictly following LSeg trained on PASCAL-5$^i$ with only 15 categories and fewer images.
> Compared with our method trained on COCO-Stuff and PASCAL VOC-15 in Table 1 in the general response, the performance increased dramatically.
>
>
>
> > **2. the proposed approach's performance when class names are provided as part of the inputs**
>
> Good suggestions!
> Without making modifications of our model,
> we design a simple strategy for class name alone as input (cls) and class name as part of the inputs (cls+attr). \
> (1) For "cls", we simply repeat the class name tokens as the same number of attribute tokens, and send into our model. \
> (2) For "cls+attr", we append class name token as one of the attribute token, then send into our model.
>
> As shown in Table 6 below,
> when we send class name only, our model is comparable with the SOTA, even if it is not intentionally designed for inputting class names.
> Appending class name with attributes has further greatly facilitated the advancement of performance, and our model fully demonstrated its capability.
> This emphasizes that attributes can effectively supplement missing information and address incompleteness of the class name input.
> With this potential of practical functionality., we firmly believe that our model can make significant contributions to the research community.
>
> *Table 6: Performance of combining class names and attributes as inputs on PAS-20. All other methods is under "direct aggregation" strategy when possible.*
> |   Model   | Backbone | Training data | mIoU (cls) | mIoU (cls+attr) | $\Delta$mIoU |
> | :-------: | :------: | :-----------: | :--------: | :-------------: | :----------: |
> |   LSeg    |  RN101   | PASCAL VOC-15 |    47.4    |      48.3       |     0.9      |
> | Fusioner  |  RN101   | PASCAL VOC-15 |    46.4    |      47.4       |     1.0      |
> | Zegformer |  RN101   |  COCO-Stuff   |    86.2    |      87.1       |     0.9      |
> |   OVSeg   |  RN101   |  COCO-Stuff   |    92.6    |      93.2       |     0.6      |
> |  CAT-Seg  |  RN101   |  COCO-Stuff   |    93.7    |      94.1       |     0.4      |
> | **Ours**  |  RN101   | PASCAL VOC-15 |    50.0    |      51.5       |   **1.5**    |
> | **Ours**  |  RN101   |  COCO-Stuff   |    93.3    |      94.6       |   **1.3**    |
>
>
> > **3. it would be beneficial to know if the proposed approach can handle hybrid inputs**
>
> We agree with reviewer that provide partial or general class names along with descriptions may enabling more realistic scenarios.
> And we are happy to say that our model are able to handle hybrid inputs, such as rough class names (e.g., "dog") combined with additional textual descriptions (e.g., "furry or smooth coat", "four-legged", "wet nose", etc.).
> This can be demonstrate in Table 6 above, with settings denote as "cls+attr".
>
> It is worth noting that although breed (e.g., "corgi") can be seen as one of the attributes, our model can handle more general descriptions, for example, colors, shapes, object parts, material, etc., which largely increased the flexibility and practicality.

---

> > ### Comment · Reviewer_mZMs · 2023-08-15
> >
> > I appreciate the authors for their efforts in addressing my inquiries. The majority of my concerns have been satisfactorily resolved, leading me to recommend the acceptance of this paper.
> >
> > Regarding the authors' explanation for the perception of "modest improvement" in their method, as indicated by the quote "Most of the latest works are trained on a larger dataset COCO-Stuff with more classes and more images," I do urge the authors to consider conducting experiments on a larger dataset and with a larger model. This exploration could shed light on the impact of both data size and model scale, which I think intriguing.

---

> > > ### Author Response · Authors · 2023-08-21
> > >
> > > We appreciate your support for our work and the positive comments.
> > > We are glad that our feedback addressed your concerns.
> > > Here is our response to the remaining concerns:
> > >
> > > > **Experiments on a larger dataset and with a larger model.**
> > >
> > > We thank your suggestion and agree that this exploration could provide interesting insights into the impact of both data size and model scale.
> > > As suggested, we conducted experiments on a larger dataset (COCO-Stuff) and with a larger model (ViT-L), following the recent open-vocabulary literature.
> > >
> > > PASCAL VOC contains 20 categories with total 1464 images for training, and PASCAL VOC-15 refers to its 4-fold cross-validation (15 categories for training).
> > > COCO-Stuff contains 171 categories with total 118k images for training, which is greatly larger than PASCAL VOC.
> > > For backbones, we also use ViT-L, which is a larger model than RN101.
> > >
> > > The results are shown in Table 10 below.
> > > Using a larger dataset and a larger model can indeed improve performance.
> > > In particular, when the dataset is larger, the improvement can be more significant.
> > >
> > > *Table 10: Performance of PAS-20 on a larger dataset and a larger model.*
> > > |                         | PASCAL VOC-15 | COCO-Stuff | $\Delta$mIoU(Dataset) |
> > > | :---------------------: | :-----------: | :--------: | :-------------------: |
> > > |          RN101          |     49.1      |    91.6    |         42.5          |
> > > |          ViT-L          |     56.4      |  **94.1**  |         37.7          |
> > > | **$\Delta$mIoU(Model)** |      7.3      |    2.5     |           -           |

---

### Official Review · Reviewer_fEW5 · 2023-07-10

**Soundness:** 3 good
**Presentation:** 3 good
**Contribution:** 3 good
**Rating:** 6
**Confidence:** 4

**Summary:**

Previous work does not address the issue of ambiguity or potentially incomplete textual categories and directly incorporates them into vision-language models without any post-processing. This approach hinders the text encoder's ability to fully comprehend the category. To overcome this limitation and provide comprehensive categorical information, the author suggests decomposing the category into more descriptive texts, such as attributes, using large language models (or human-designed prompts). Subsequently, the decomposed texts are aggregated using the proposed method.

**Strengths:**

1. Sound motivation.
The core idea of distinguishing categories by decomposing them into attributes and aggregating them sounds promising.

2. Suggesting a new dataset to support the motivation.


**Weaknesses:**

1. Missing citations.
The authors should include citations to recent open-vocabulary semantic segmentation literature and make a comparison with them [2, 3, 4].

2. Although the motivation is good, the experiment is insufficient.
The benchmark conducted by the authors is not sufficient to provide a complete comparison with recent open-vocabulary segmentation works. The authors conducted experiments on benchmarks widely used in few-shot segmentation literature, rather than in open-vocabulary semantic segmentation. To the best of my knowledge, only two papers have utilized this benchmark in an open-vocabulary setup (LSeg, Fusioner [1]). This choice prevents a comparison with recent state-of-the-art open-vocabulary methods such as Zegformer [2], OVSeg [3], or CAT-Seg [4].

3. Lack of analysis of the motivation for **decomposition**.
The provided analyses primarily focus on demonstrating the effectiveness of the aggregation modules, rather than specifically proving the effectiveness of the decomposition approach. It would be beneficial for the author to include ablation studies and analyses of the **decomposition** within the main paper.



**Questions:**

The authors chose to aggregate visual and attribute information together in one model. However, considering the pre-training nature of vision-language models, the most naive way to achieve the same goal would be to concatenate all the attributes into a sequence and input them into a text encoder. The resultant text embedding can then be used as a classifier, which can be applied to existing methods such as Zegformer or OVSeg. I'm curious about the performance of this naive method, which is the most easiest way to confirm the validity of the motivation of decomposition.

**Limitations:**

The motivation seems promising; however, I think this motivation can be applied to all of the existing methods in open-vocabulary literature. In other words, the method the author suggests does not seem to emphasize the advantage of the motivation but rather has high similarity to previous work (Fusioner [1]). This paper would be much stronger if the author provided an analysis regarding the motivation instead of focusing solely on the technical design.

[1] Ma et. al., "Open-vocabulary Semantic Segmentation with Frozen Vision-Language Models", BMVC'22
[2] Ding et. al., "Decoupling Zero-Shot Semantic Segmentation", CVPR'22
[3] Liang et. al., "Open-Vocabulary Semantic Segmentation with Mask-adapted CLIP", CVPR'23
[4] Cho et. al, "CAT-Seg: Cost Aggregation for Open-Vocabulary Semantic Segmentation", arXiv'23

---

> ### Author Rebuttal · Authors · 2023-08-10
>
>
> ## Reviewer fEW5
>
> Thank you for your appreciations on the novelty, motivation, and results of the idea, as well as the detailed and constructive review.
> We address individual questions and concerns below.
>
> > **1. Missing citations**
>
> Thanks for pointing out.
> We will add these literatures in related works, and the comparisons such as Table 1, 2 in general response will also be added into the revised paper.
>
>
> > **2. Comparison with recent state-of-the-art open-vocabulary methods such as Zegformer, OVSeg, or CAT-Seg**
>
> Thanks for acknowledging our motivation and the constructive suggestions.
> As reviewer suggested, we have added comparisons with recent SOTA methods in Table 1, 2 in the general response.
> Note that, the mentioned papers are evaluated on ADE20K, PASCAL VOC, and PASCAL-Context datasets, and they are trained on a larger dataset COCO-Stuff with more classes and more images.
> Considering the burden of annotating the attributes for all datasets during rebuttal, we here annotate attributes of some representative dataset: COCO-Stuff for training, and PASCAL-Context (PC-59) and PASCAL VOC (PAS-20) for evaluation.
>
>
> > **3. Ablation studies of the decomposition**
>
> Good suggestions!
> We conducted two types of ablations on Fanatic Beasts:
>
> (1) Number: varying the number of attributes used for class decomposition (Table 4).
> Increasing the number of attributes resulted in improved performance.
>
> (2) Diversity: exploring different types of decomposed attributes (Table 5). We roughly categorize the decomposed attributes into four types: color, shape, parts, and others, and maintain a consistent total number of inputs.
> All types of attributes contribute to the overall performance, highlighting the significance of attribute diversity.
>
>
> *Table 4: Ablation on the number of the attributes for the class decomposed into.*
> | Number of decomposed attr |  5   |  10  |  15  |
> | :-----------------------: | :--: | :--: | :--: |
> |           mIoU            | 47.8 | 50.9 | 52.3 |
>
>
>
> *Table 5: Ablation on different types of decomposed attributes.*
> | Color | Shape | Parts | Others | mIoU |
> | :---: | :---: | :---: | :----: | :--: |
> |   √   |       |       |        | 30.8 |
> |   √   |   √   |       |        | 42.5 |
> |   √   |   √   |   √   |        | 51.0 |
> |   √   |   √   |   √   |   √    | 52.3 |
>
>
>
>
> > **4. The performance of a naive method which can confirm the validity of the motivation of decomposition**
>
> We totally agree with the reviewer that "the most naive way to achieve the same goal would be to concatenate all the attributes into a sequence and input them into a text encoder".
> In our work, we have referred to this approach as "direct aggregation," as indicated in Figure 3.(a) of the original paper.
>
> To compare various aggregation strategies, we had conducted an ablation in Sec.4.2, *i.e.*, Table 3, in the original paper. This ablation mainly focuses on our method and LSeg on PASCAL.
> Additionally, as suggested by the reviewer,
> we extended experiments on "direct aggregation" to existing SOTA methods such as Zegformer or OVSeg, on multiple benchmarks.
> The results of these experiments are presented in Table 1 and 2 of the general response.
>
> Overall, all the results demonstrate that, "direct aggregation" strategy is generally effective, especially in real-world scenarios where neologisms and unnameable objects may occur.
> Our proposed hierarchical aggregation achieved superior results.
> This definitely confirm the validity of the motivation.
>
>
>
> > **5. The motivation can be applied to all of the existing methods in open-vocabulary literature**
>
>
> We agree with the reviewer that the motivation can be applied to all of the existing methods in open-vocabulary literature.
>
> The motivation of our work is to first decompose class names into attributes,
> which can solve the issue of ambiguity, neologisms and unnameability, then aggregate them for segmentations.
> This can be equipped on all recent SOTA methods on-the-shelf.
> As shown in Table 2 in the general response, the proposed decomposition-aggregation ideology successfully validate its effectiveness in challenging real world scenarios that may encounter various situations like ambiguity, neologism and unnameability.

---

> > ### Comment · Reviewer_fEW5 · 2023-08-12
> >
> > Thank you for your response!
> > The experiments all look really interesting, and I believe the results should be included in the main paper.
> > I hope to see how future work will further develop this idea.

---

> > > ### Author Response · Authors · 2023-08-17
> > >
> > > Thank you for the response and appreciation!
> > > All experiments and results will certainly be appended in the revised paper.
> > >
> > > We also value your suggestion.
> > > Considering the value of attributes, some possible directions for future work include:
> > >
> > > 1. More comprehensive visual tasks. To extend our motivation and framework to various tasks, such as instance segmentation, object detection, and image retrieval, etc.
> > > 2. Robust strategies on attribute noise. To design more efficient and effective attribute decomposition and aggregation methods that can handle large-scale and noisy attribute descriptions, such as using attention mechanisms or graph neural networks.
> > > 3. Hierarchical cross-modal alignment. To investigate the hierarchical structure within attributes, as well as aligning them with different levels of visual features. A potential approach could involve leveraging hyperbolic geometry, by embedding them in a hyperbolic space.
> > >
> > > In conclusion, these future works aim to guide the community towards more practical directions.
> > > We hope these advancements will better serve and enhance our daily lives.

---

### Official Review · Reviewer_cgtT · 2023-07-17

**Soundness:** 3 good
**Presentation:** 3 good
**Contribution:** 2 fair
**Rating:** 6
**Confidence:** 4

**Summary:**

This paper proposes a framework for decomposing class names into attributes for open vocabulary semantic segmentation. The authors obtain attributes from LLMs or hand-curate for unheard-of classes, then aggregates these attributes for obtaining a general descriptor for each classes. Moreover, the authors collect a dataset to demonstrate their effectiveness on addressing neologisms and unnameability for OVSS.

**Strengths:**

1. The motivation for the paper, addressing the problems existing in pre-trained VLM-based paradigm for OVSS seems solid.
2. Evaluating the proposed method on a newly collected dataset, consisted of imaginary creatures, is an interesting method for evaluating how models can handle "neologism" and "unnameability" addressed in the paper.

**Weaknesses:**

1. Insufficient evidence for supporting the work
- The authors evaluate with COCO-20i[19] and PASCAL-5i[10, 13], which have brief, but well-known classes. These classes barely seem to suffer problems such as neologism or unnameability, and does not illustrate cases where ambiguity might be an issue.
- Moreover, the collected attributes for COCO and PASCAL are not present in the paper or the supplementary materials, making it harder to validate how these attributes may contribute to the quality of the output segments.

2. Weaknesses in quantitative comparison and ablations
- The current comparison with LSeg[17] does not seem to sufficiently provide evidence for the core contribution of the work of attribute decomposition-aggregation, as LSeg[17] does not improve with provided attributes.
The improvements may seem to come form the fusion layers, which seems to beg similarity with Fusioner[21]. Comparison with Fusioner[21] would better illustrate the significance of this work, as it would be able to solely ablate the proposed attribute aggregation.

3. Flaws for addressing the presented problem
- If the class names are ambiguous by their own, how are LLMs capable of distinguishing between this ambiguity? For example, how would LLMs be able to distinguish between the bird "crane" and the machine "crane" with the following prompt "Describe what a {crane} looks like in the image.", without further elaborating the context?
- The claim that "segmentation through attributes is a more challenging task"(L.281) seems to be a bold claim, as the authors obtain the attributes as additional information from the class names, with the help from LLMs or human annotation.

**Questions:**

1. The collected dataset, "Fantastic Beasts" is collected from a movie series, which may raise concerns regarding copyright issues. Can the  authors include discussions, or clarify how the copyright may not be of an issue?

**Limitations:**

Limitations regarding the collection of the dataset should be further discussed.

---

> ### Author Rebuttal · Authors · 2023-08-10
>
> Thank you for acknowledging our novelty and motivation, we address your kind concerns below.
>
> > **1.1. COCO/PASCAL have well-known classes without neologism etc**
>
> There may be some misunderstandings. We admit COCO/PASCAL have well-known classes. However, the aim of using them are **_not_** intended for neologism or unnameability. Instead, as we pioneer the segmentation exploration via attribute, there's no existing benchmark to meet our requirements. We hence annotate attributes on widespread COCO/PASCAL.
> 1. For training, these datasets give *representative simulation environments*, i.e., image-mask-attributes pairs with well-defined 4 folds for cross validation. Models training on this dataset can successfully adapt to real-world scenarios, such as Fantastic Beasts, with neologism or unnameability. This proves the rationality of this environment.
> 2. For evaluation, COCO/PASCAL are *standard benchmarks* to show our effectiveness(admit by R-fEW5&mZMs). Besides, they also allow us to easier assess the performance of other established baselines, facilitating understanding.
>
> Furthermore, as suggested, we also included datasets used in SOTA of OVSS for additional comparison (see Table 1,2).
>
> > **1.2. Attributes for COCO/PASCAL are not shown**
>
> We apologize for the inconvenience. Limited by page, we only provide some attributes of PASCAL. Whole list attached in supplementary of the revised version.
> ```
> bird: two wings,two legs,a beak,feathers,a tail,bright colors,a curved neck,small eyes,a pointed head,a small body,a pointed beak,webbed feet
> ```
>
> > **2.1. The comparison with LSeg does not seem to be sufficient, as LSeg does not improve with attributes**
>
> We believe there are some misunderstandings.
> 1. For common classes, our intention is not to rely solely on attributes inputs to surpass the SOTA that uses class name inputs. As clarified in the general response, attribute-based segmentation presents greater challenges.
> Without considering the crucial information of class names and comparing it to class name-based methods would be unfair.
> 2. A fair comparison is to use class name with attributes as input for the usual categories (see Table 6 of R-mZMs), to show attributes input can supplement incomplete class information.
> 3. When faced with real-world scenarios that have possibility of neologisms or unnameability, attribute-based methods are significantly better than that with class name inputs (see Table 2 of the general response). This directly demonstrates the effectiveness of our idea.
> 4. To our knowledge, we are the first to work on attribute-based segmentation. Our initial successful exploration aims to lead the community's attention towards this practical scenario, thus promoting further development.
>
> > **2.2. Improvements come from the fusion layers? Comparison with Fusioner**
>
> We should like to clarify that improvements do not mainly come from the fusion layers.
> Table 3 below gives comparisons with Fusioner. Results show the gain mainly comes from the clustering module. The fusion layer only gives 1.2% mIoU gains on average, compared with a greater 4.3% of clustering module. Besides, for more comprehensive comparisons, please refer to Table 1,2 in the general response.
>
> *Table 3: Ablation of fusion layer and aggregation module on PASCAL-5$^i$.*
> Model (Components)|Settings|5$^0$|5$^1$|5$^2$|5$^3$|mIoU|$\Delta$w/"full"
> :-:|:-:|:-:|:-:|:-:|:-:|:-:|:-:
> Fusioner|cls name|46.8|56.0|42.2|40.7|46.4|-
> |
> Ours (full)|attr|52.9|55.3|45.0|43.1|49.1|-
> Ours (w/o fusion layer)|attr|52.0|54.6|43.1|41.9|47.9|$\downarrow$ 1.2
> Ours (w/o clustering module)|attr|48.8|51.4|39.7|39.2|44.8|$\downarrow$ 4.3
>
> And although both work use fusion-like layers, ours is very different from Fusioner:
> 1. Fusioner belongs to class-name-based segmentation method like LSeg, while we are attribute-based without class name.
> 2. Fusioner designes fusion layers to merge image and text features, while we aim to information propagation, i.e., exchanging visual information with grouped attribute features and learnable cluster tokens for each stage.
>
> > **3.1. LLMs cannot distinguish ambiguous names**
>
> We agree, LLMs cannot distinguish ambiguity without additional inputs, and human is required at this point. However, the intention of this work is *not* to eliminate ambiguity *for LLM*s. As a pioneer for attribute segmentation, our main challenge lies in building standard datasets with minimal cost. Using LLMs to assist proves to be an efficient approach, particularly when dealing with many classes. Human annotators only need to manually filter and verify generated attributes, which significantly reduces costs.
>
> > **3.2. "segmentation via attributes is more challenging" seems to be bold**
>
> We believe that there are some misunderstandings. Please see the general response for detailed clarification.
> Attribute segmentation is more challenging can also be observed from Table 1 in the original paper (LSeg (attr) vs LSeg (cls name)), especially for common classes. Nevertheless, for common classes, this issue can be solved by combining class names and attributes as input. Further details are shown in Table 6 (R-mZMs).
>
> > **4. computational concerns**
>
> Computational is not a concern, and it's not the primary focus.
> We transform OVSS into multiple binary segmentation tasks, rather than treating it as a multi-class segmentation with inputting multi-class attributes simultaneously.
> 1. The computation time may demonstrate a linear relationship with class number. This is acceptable for our pioneer exploration.
> 2. The GPU memory is significantly reduced, since the input sequence is much shorter, only 20 tokens for all classes compared to 20*N tokens for N classes.
>
> Overall, our key goal is to show the value of attribute decomposition-aggregation. Futures work may explore ways to reduce computational time.
>
> > **5. Copyright concern**
> We believe it is "fair use" and is not an infringement of copyright. We will contact the movie company to validate.

---

> > ### Comment · Reviewer_cgtT · 2023-08-12
> >
> > I thank the authors for the detailed response. Additional experiments does answer a lot of my concerns, and I think they should definitely be in the paper. I also agree that COCO/PASCAL are indeed standard benchmarks. I have adjusted my initial rating, but my remaining concerns are:
> >
> > 1. Since "textual ambiguity" requires human annotation, this seems like it can be easily handled by other methods by just specifying "crane bird" or "crane machine". How can the proposed method be more effective or efficient in this perspective?
> >
> > 2. Comparison with SOTA works. From current results, the authors use "direct aggregation" when evaluating with SOTA works, where all the attributes are concatenated to a single sentence for CLIP. The problem is that CLIP can struggle with this long, detailed description and might better perform with short, distinctive descriptions, hence current evaluation seems a bit unfair.
> > Since the use of human annotation seems inevitable, can the authors evaluate SOTA works on Fantastic beasts with a **short description**, for example like "small black creature"?
> >
> > 3. Further investigation of "cls+attr". I think the results from Table 6 are fascinating, and can be joined with Table 7. Since a more common case would be having the class name, "cls+attr" seems to be a more reasonable and general case compared to only having attributes. In this perspective, can the authors show results when having **wrong classnames**, where the attributes can act as a **fallback** for addressing neologism or unnameability?
> >
> > Thanks.

---

> > > ### Author Response · Authors · 2023-08-17
> > >
> > > We appreciate the reviewer for acknowledging our rebuttal, and we are pleased that our feedback addresses most of your concerns.
> > > We will definitely include all these additional experiments in the revised paper.
> > >
> > > Furthermore, we respect the responsibility of the reviewer, in providing constructive suggestions that can enhance the paper and make it even better.
> > > Here is our response to the remaining concerns:
> > >
> > > > **1. Other methods can handle this by specifying "crane bird" or "crane machine". How can the proposed method be more effective or efficient in this perspective?**
> > >
> > > We agree that other methods can handle this scenario by inputting "crane bird" or "crane machine".
> > >
> > > However, it is worth noting that "bird" and "machine" are also one of the attributes associated with concepts of a "crane".
> > > Specifying "bird" or "machine" is considered as specifying crane's attributes.
> > > These instances further align with the idea we have proposed, i.e., decomposing into attributes and aggregation, serving as compelling examples that highlight the generality of our motivation.
> > >
> > > Besides, as stated in the response to the third question of R-mZMs, our method is capable of handling "hybrid inputs".
> > > For example, it can process general class names (e.g., "dog") or specify the breed (e.g., "corgi").
> > > This can enhance the effectiveness and efficiency of our method.
> > > We can consider "crane bird/machine" to be an example that falls into this situation.
> > > As demonstrated in Table 6, our method outperforms existing methods when using hybrid inputs.
> > >
> > > > **2. CLIP can struggle with long, detailed description when concatenated, hence current evaluation seems a bit unfair. Can the authors evaluate SOTA works on Fantastic beasts with a short description?**
> > >
> > > We first clarify that after concatenation of all attributes belong to their class, none of the categories have exceeded the specified maximum sequence length for the model to process.
> > > We present statistical data, with an average length of approximately 55 for all categories, and the longest length is 75.
> > > These lengths are all well within the maximum sequence length of 77 (the CLIP model we used).
> > > So we believe that the length of the concatenated attributes is not a problem for CLIP to handle, thus the current evaluation is fair.
> > >
> > > As suggested, we also conducted experiments on Fantastic Beasts with a short attributes description.
> > > In order to ensure maximum attributes diversity such as color, shape, parts, etc., we selected *3* most distinctive attributes for each category in Fantastic Beasts (3 is significantly fewer compared to the original 15 attributes).
> > > After concatenation, the total length of these 3 attributes is not exceed 9, with an average around 6 (some attributes may contain 2 words, for example, "small, blue body, thin wings").
> > >
> > > The results are shown in Table 8 below. Our methods still demonstrate the superiority.
> > >
> > > *Table 8: Evaluation on Fantastic Beasts with a short attributes description.*
> > > Model|Backbone|Training data|mIoU(short attr)
> > > :-:|:-:|:-:|:-:
> > > LSeg|RN101|PASCAL VOC-15|40.7
> > > Fusioner|RN101|PASCAL VOC-15|41.1
> > > Zegformer|RN101|COCO-Stuff|52.4
> > > OVSeg|RN101|COCO-Stuff|54.0
> > > CAT-Seg|RN101|COCO-Stuff|54.6
> > > **Ours**|RN101|PASCAL VOC-15|47.4
> > > **Ours**|RN101|COCO-Stuff|**56.0**
> > >
> > >
> > > > **3. "cls+attr" seems to be a more reasonable and general case compared to only having attributes. In this perspective, can the authors show results when having wrong classnames, where the attributes can act as a fallback?**
> > >
> > > We agree with reviewer's suggestion where the attributes act as a fallback when having wrong class names.
> > > In real world scenarios, such situations can potentially occur where users may input incorrect class names while also providing accurate attributes.
> > >
> > > Considering this, we give experiments based on Table 6 and Table 7, and treat the wrong class name as one incorrect attribute.
> > > The wrong class name is randomly selected from the remaining categories, excluding the correct category.
> > >
> > > As shown in Table 9, inputting wrong class names will lead to extremely poor results.
> > > Since class names hold core information (mentioned in the general response), the wrong class name will result in a completely incorrect textual classifier for VLP-based methods.
> > > On the contrary, however, the attribute can provide correct information from multiple distinct or complementary perspectives, which can act as a "fallback" when the class name is wrong.
> > >
> > > *Table 9: Performance of wrong class name inputs, where attributes can act as a fallback.*
> > > Model|Backbone|Training data|mIoU(correct cls)|mIoU(wrong cls)|mIoU(wrong cls+attr)|$\Delta$w/ correct cls
> > > :-:|:-:|:-:|:-:|:-:|:-:|:-:
> > > LSeg|RN101|PASCAL VOC-15|47.4|<10|41.6|12.2%
> > > Fusioner|RN101|PASCAL VOC-15|46.4|<10|41.8|9.91%
> > > Zegformer|RN101|COCO-Stuff|86.2|<10|76.1|11.7%
> > > OVSeg|RN101|COCO-Stuff|92.6|<10|81.3|12.2%
> > > CAT-Seg|RN101|COCO-Stuff|93.7|<10|84.7|9.6%
> > > **Ours**|RN101|PASCAL VOC-15|50.0|<10|45.9|8.2%
> > > **Ours**|RN101|COCO-Stuff|93.3|<10|86.0|**7.82%**

---

> > > > ### Comment · Reviewer_cgtT · 2023-08-19
> > > >
> > > > Thank you for the additional response. I am glad to see my remaining concerns are sufficiently addressed in the provided results, and based on them, I am convinced that this paper holds interesting values for the field of open-vocabulary segmentation. I am looking forward to see the paper strengthened with the additional experiments, and hence adjust my rating accordingly.

---

> > > > > ### Author Response · Authors · 2023-08-21
> > > > >
> > > > > Thank you for your detailed review and taking time to provide feedback on our paper!
> > > > > We are glad to hear that our additional response has sufficiently addressed all your concerns.
> > > > > We are excited to continue our work on open-vocabulary segmentation and will incorporate the additional experiments to further strengthen our paper.
> > > > > Thank you again for your valuable comments.

---

### Author Rebuttal · Authors · 2023-08-10

## General Response and Clarifications

We kindly thank the reviewers for their detailed reading, and considerate feedback. We are also grateful of the appreciations by the reviewers in these aspects:

(1) **Novel and well-motivated idea**: "motivation seems solid" (R-cgtT); "sound motivation and promising" (R-fEW5); "novel framework, limitations of existing works are critical" (R-mZMs); "idea is straightforward and easy to understand" (R-PB8W).

(2) **Significance of the newly collected dataset**: "interesting for evaluation" (R-cgtT); "support the motivation" (R-fEW5).

(3) **Superior performance and thorough experiments**: "flexibility and accommodates different scenarios" (R-mZMs); "superior performance" (R-mZMs); "thorough ablation"(R-mZMs); "the proposed method is effective" (R-PB8W).

Based on these acknowledgements, reviewers suggest that further empirical results will make this paper stronger, serving to highlight its strengths and clarify motivations. We agree, and have made every effort to execute the majority of the suggested experiments within the limited time of the rebuttal phase.

We would like to offer these general concerns and clarifications below.

### 1. Additional Experiments

#### 1.1 Comparison on recent SOTA methods on more benchmarks

Recent SOTA methods like Zegformer, OVSeg and CAT-Seg are trained on a larger dataset COCO-Stuff with more classes and more images.
Following them, we select some representative datasets PASCAL-Context (PC-59) and PASCAL VOC (PAS-20) and annotating the attributes them for evaluation.

The results are in Table 1 below. Our method still demonstrates superiority.

*Table 1: Comparison on recent SOTA methods on more benchmarks. All other methods is under "direct aggregation" strategy.*
|Model|Settings|Backbone|Training data|PC-59|PAS-20
|:-:|:-:|:-:|:-:|:-:|:-:
|LSeg|attr|RN101|PASCAL VOC-15|24.2|44.0
|Fusioner|attr|RN101|PASCAL VOC-15|25.2|44.3
|Zegformer|attr|RN101|COCO-Stuff|39.0|83.7
|OVSeg|attr|RN101|COCO-Stuff|51.2|90.3
|CAT-Seg|attr|RN101|COCO-Stuff|53.6|90.9
|**Ours**|attr|RN101|PASCAL VOC-15|29.1|49.1
|**Ours**|attr|RN101|COCO-Stuff|**56.3**|**91.6**

#### 1.2 Apply decomposition-aggregation for existing SOTA methods in real-world scenarios

We demonstrate the advantage of the motivation by showing that the proposed decomposition-aggregation ideology can be applied to existing methods in open-vocabulary literature.
We take Fantastic Beasts as an outstanding representative of the real-world that may encounter various situations like ambiguity, neologism and unnameability.

As shown in Table 2 below, the existing SOTA methods are not able to handle the real-world scenarios when taking class name as input (mIoU (cls)).
However, when decomposing into attributes and then aggregating them,
the performance of these methods remarkably increase, showing significant gains with robust performance (mIoU (attr)).
This demonstrates the universality and effectiveness of our motivation.

*Table 2: Effectiveness of our motivation in real-world scenarios. All other methods is under "direct aggregation" strategy when possible.*
Model|Backbone|Training data|mIoU(cls)|mIoU(attr)
:-:|:-:|:-:|:-:|:-:
LSeg|RN101|PASCAL VOC-15|<10|46.0
Fusioner|RN101|PASCAL VOC-15|<10|46.1
Zegformer|RN101|COCO-Stuff|<20|55.7
OVSeg|RN101|COCO-Stuff|<20|58.1
CAT-Seg|RN101|COCO-Stuff|<20|59.4
**Ours**|RN101|PASCAL VOC-15|-|52.3
**Ours**|RN101|COCO-Stuff|-|**61.9**

### 2. Is segmentation by only attributes more challenging than by class names under the current VLP diagram?

Yes, segmentation by only attributes is *indeed* more difficult than by class names as it poses additional challenges.

(1) Defining attributes: The definition of attributes is crucial and requires careful consideration.

(2) Noise in attributes: Developing effective methods to resist noise in attribute is essential for accurate results.

(3) How to aggregating complementary information: The critical aspect lies in effectively aggregating the complementary information provided by different attributes.

(4) Attributes describe the categories from *indirect* perspectives:
As the fundamental aim of VLP is to align image modality with its textual *class name*,
it is undeniable that class names hold core information.
Recent OVSS are all based on VLP, for example, they use CLIP as classifiers, which are inherently bound by this characteristic.
In contrast to class names, which provide the most direct information, attributes describe categories from indirect perspectives, adding to the challenge when using only attributes for segmentation.

### 3. Why do we opt for attribute inputs instead of utilizing class names, since segmentation by attributes are more challenging?

Because of the inherent flaws of VLP, there may be situations where it is not possible to input the category name.
For example, ambiguity, unknown, or unnameable category names.
Input attributes is capable of effectively addressing these issues, by making up for missing context information, transforming into known attributes or describing in a more detailed and accurate way.
This is also the motivation of this work, and Table 2 in the original paper and Table 2 above demonstrate this point.

For common classes such as in PASCAL and COCO, instead of input attributes only, as suggested by R-mZMs, we give a simple strategy for combining class names and attributes as input.
Despite not specifically designed for mixed input, our method can still handle this situation.
This further enhanced its practicality in real life, please refer to the response for R-mZMs for details.

### 4. Conclusion

To summarize, we thank the reviewers for their careful feedback and additional suggestions for evaluation, which will make our paper significantly stronger.
We look forward to further discussion, and are happy to answer any questions that might arise.

---

### Decision · Program_Chairs · 2023-09-21

**Decision:**

Accept (poster)

**Comment:**

This paper has been reviewed by four experts in the field. After rebuttal and discussion, all reviewers acknowledged contributions of this paper, including technical novelty and thorough experiments. The ACs agreed with the reviewers.